# Characterizing Generalization under Out-Of-Distribution Shifts in Deep Metric Learning

**Timo Milbich**[*]
LMU Munich & IWR, Heidelberg University
timo.milbich@iwr.uni-heidelberg.de

**Karsten Roth**[*,x]
IWR, Heidelberg University
karsten.rh1@gmail.com

**Samarth Sinha**
University of Toronto, Vector
sinhasam@fb.com

**Ludwig Schmidt**
University of Washington
schmidt@cs.uw.edu

**Marzyeh Ghassemi**[†]
MIT, University of Toronto, Vector
mghassem@mit.edu

**Björn Ommer**[†]
LMU Munich & IWR, Heidelberg University
ommer@uni-heidelberg.de

## Abstract

Deep Metric Learning (DML) aims to find representations suitable for zero-shot transfer to *a priori* unknown test distributions. However, common evaluation protocols only test a single, fixed data split in which train and test classes are assigned randomly. More realistic evaluations should consider a broad spectrum of distribution shifts with potentially varying degree and difficulty. In this work, we systematically construct train-test splits of increasing difficulty and present the ***ooDML*** benchmark to characterize generalization under ***o***ut-***o***f-distribution shifts in ***DML***. *ooDML* is designed to probe the generalization performance on much more challenging, diverse train-to-test distribution shifts. Based on our new benchmark, we conduct a thorough empirical analysis of state-of-the-art DML methods. We find that while generalization tends to consistently degrade with difficulty, some methods are better at retaining performance as the distribution shift increases. Finally, we propose few-shot DML as an efficient way to consistently improve generalization in response to unknown test shifts presented in *ooDML*[1].

## 1 Introduction

Image representations that generalize well are the foundation of numerous computer vision tasks, such as image and video retrieval [61, 71, 54, 38, 1], face (re-)identification [57, 34, 8] and image classification [65, 4, 19, 40, 37]. Ideally, these representations should not only capture data within the training distribution, but also transfer to new, out-of-distribution (OOD) data. However, in practice, achieving effective OOD generalization is more challenging than in-distribution [28, 12, 21, 49, 31, 55]. In the case of zero-shot generalization, where train and test classes are completely distinct, Deep Metric Learning (DML) is used to learn metric representation spaces that capture and transfer visual similarity to unseen classes, constituting *a priori* unknown test distributions with unspecified shift. To approximate such a setting, current DML benchmarks use *single, predefined and fixed* data splits of disjoint train and test classes, which are assigned arbitrarily [71, 8, 61, 24, 11, 33, 74, 51, 54, 42, 26, 64, 58]. This means that *(i)* generalization is only evaluated on a fixed problem difficulty, *(ii)*

---

[1]Code available here: https://github.com/CompVis/Characterizing_Generalization_in_DML

[*] Equal contribution, alphabetical order, [†] equal supervision, [x] now at University of Tuebingen.

generalization difficulty is only implicitly defined by the arbitrary data split, *(iii)* the distribution shift is not measured and *(iv)* cannot be not changed. As a result, proposed models can overfit to these singular evaluation settings, which puts into question the true zero-shot generalization capabilities of proposed DML models.

In this work, we first construct a new benchmark *ooDML* to characterize *g*eneralization under *out-of*-distribution shifts in *DML*. We systematically build *ooDML* as a comprehensive benchmark for evaluating OOD generalization in changing zero-shot learning settings which covers a much larger variety of zero-shot transfer learning scenarios potentially encountered in practice. We systematically construct training and testing data splits of increasing difficulty as measured by their Frechet-Inception Distance [23] and extensively evaluate the performance of current DML approaches.

Our experiments reveal that the standard evaluation splits are often close to i.i.d. evaluation settings. In contrast, our novel benchmark continually evaluates models on significantly harder learning problems, providing a more complete perspective into OOD generalization in DML. Second, we perform a large-scale study of representative DML methods on *ooDML*, and study the actual benefit of underlying regularizations such as self-supervision [38], knowledge distillation [53], adversarial regularization [59] and specialized objective functions [71, 70, 8, 26, 54]. We find that conceptual differences between DML approaches play a more significant role as the distribution shift to the test split becomes harder.
Finally, we present a study on few-shot DML as a simple extension to achieve systematic and consistent OOD generalization. As the transfer learning problem becomes harder, even very little in-domain knowledge effectively helps to adjust learned metric representation spaces to novel test distributions. We publish our code and train-test splits on three established benchmark sets, CUB200-2011 [68], CARS196 [30] and Stanford Online Products (SOP) [43]. Similarly, we provide training and evaluation episodes for further research into few-shot DML. Overall, our contributions can be summarized as:

- Proposing the *ooDML* benchmark to create a set of more realistic train-test splits that evaluate DML generalization capabilities under increasingly more difficult zero-shot learning tasks.
- Analyzing the current DML method landscape under *ooDML* to characterize benefits and drawbacks of different conceptual approaches to DML.
- Introducing and examining few-shot DML as a potential remedy for systematically improved OOD generalization, especially when moving to larger train-test distribution shifts.

## 2 Related Work

DML has become essential for many applications, especially in zero-shot image and video retrieval [61, 71, 51, 24, 1, 36]. Proposed approaches most commonly rely on a surrogate ranking task over tuples during training [62], ranging from simple pairs [17] and triplets [57] to higher-order quadruplets [5] and more generic n-tuples [61, 43, 22, 70]. These ranking tasks can also leverage additional context such as geometrical embedding structures [69, 8]. However, due to the exponentially increased complexity of tuple sampling spaces, these methods are usually also combined with tuple sampling objectives, relying on predefined or learned heuristics to avoid training over tuples that are too easy or too hard [57, 72] or reducing tuple redudancy encountered during training [71, 15, 18, 52]. More recent work has tackled sampling complexity through the usage of proxy-representations utilized as sample stand-ins during training, following a NCA [16] objective [41, 26, 64], leveraging softmax-style training through class-proxies [8, 73] or simulating intraclass structures [46].

Unfortunately, the true benefit of these proposed objectives has been put into question recently, with [54] and [42] highlighting high levels of performance saturation of these discriminative DML objectives on default benchmark splits under fair comparison. Instead, orthogonal work extending the standard DML training paradigm through multi-task approaches [56, 51, 39], boosting [44, 45], attention [27], sample generation [11, 33, 74], multi-feature learning [38] or self-distillation [53] have shown more promise with strong relative improvements under fair comparison [54, 38], however still only in single split benchmark settings. It thus remains unclear how well these methods generalize in more realistic settings [28] under potentially much more challenging, different train-to-test distribution shifts, which we investigate in this work.

Table 1: FID scores between i.i.d. subsampled training and test sets in comparison to FID scores measured on default splits used in standard DML evaluation protocols. As can be seen, the train-test distribution shift of two out of three benchmarks are actually close i.i.d. settings, in particular when compared to the train-test splits evaluated in Fig. 1 reaching scores over 200.

| Dataset $\rightarrow$ | CUB | CARS | SOP |
|---|---|---|---|
| **Default** - different classes train/test | 52.62 | 8.59 | 3.43 |
| **i.i.d.** - same classes train/test | $4.87 \pm 0.05$ | $2.33 \pm 0.03$ | $0.98 \pm 0.01$ |

## 3 *ooDML*: Constructing a Benchmark for OOD Generalization in DML

An image representation $\phi(x)$ learned on samples $x \in \mathcal{X}_{\text{train}}$ drawn from some training distribution generalizes well if can transfer to test data $\mathcal{X}_{\text{test}}$ that are not observed during training. In the particular case of OOD generalization, the learned representation $\phi$ is supposed to transfer to samples $\mathcal{X}_{\text{test}}$ which are not independently and identically distributed (i.i.d.) to $\mathcal{X}_{\text{train}}$. A successful approach to learning such representations is DML, which is evaluated for the special case of zero-shot generalization, i.e. the transfer of $\phi$ to distributions of unknown classes [57, 71, 24, 8, 54, 42]. DML models aim to learn an embedding $\phi$ mapping datapoints $x$ into an embedding space $\Phi$, which allows to measure similarity between $x_i$ and $x_j$ as $g(\phi(x_i), \phi(x_j))$. Typically, $g$ is a predefined metric, such as the Euclidean or Cosine distance and $\phi$ is parameterized by a deep neural network.

In realistic zero-shot learning scenarios, test distributions are not specified a priori. Thus, their respective distribution shifts relative to the training, which indicates the difficulty of the transfer learning problem, is unknown as well. To determine the generalization capabilities of $\phi$, we would ideally measure its performance on different test distributions covering a large spectrum of distribution shifts, which we will also refer to as "problem difficulties" in this work. Unfortunately, standard evaluation protocols test the generalization of $\phi$ on a *single and fixed* train-test data split of predetermined difficulty, hence only allow for limited conclusions about zero-shot generalization.

To thoroughly assess and compare zero-shot generalization of DML models, we aim to build an evaluation protocol that resembles the undetermined nature of the transfer learning problem. In order to achieve this, we need to be able to *change*, *measure* and *control* the difficulty of train-test data splits. To this end, we present an approach to construct multiple train-test splits of measurably increasing difficulty to investigate *o*ut-*o*f-distribution generalization in **DML**, which make up the *ooDML* benchmark. Our generated train-test splits resort to the established DML benchmark sets, and are subsequently used in Sec. 4 to thoroughly analyze the current state-of-the-art in DML. For future research, this approach is also easily applicable to other datasets and transfer learning problems.

### 3.1 Measuring the gap between train and test distributions

To create our train-test data splits, we need a way of measuring the distance between image datasets. This is a difficult task due to high dimensionality and natural noise in the images. Recently, Frechet Inception Distance (FID) [23] was proposed to measure the distance between two image distributions by using the neural embeddings of an Inception-v3 network trained for classification on the ImageNet dataset. FID assumes that the embeddings of the penultimate layer follow a Gaussian distribution, with a given mean $\mu_{\mathcal{X}}$ and covariance $\Sigma_{\mathcal{X}}$ for a distribution of images $\mathcal{X}$. The FID between two data distributions $\mathcal{X}_1$ and $\mathcal{X}_2$ is defined as:

$$d(\mathcal{X}_1, \mathcal{X}_2) \triangleq \|\mu_{\mathcal{X}_1} - \mu_{\mathcal{X}_2}\|_2^2 + \text{Tr}(\Sigma_{\mathcal{X}_1} + \Sigma_{\mathcal{X}_2} - 2(\Sigma_{\mathcal{X}_1}\Sigma_{\mathcal{X}_2})^{\frac{1}{2}}) , \qquad (1)$$

In this paper, instead of the Inception network, we use the embeddings of a ResNet-50 classifier (Frechet *ResNet* Distance) for consistency with most DML studies (see e.g. [71, 64, 26, 56, 51, 38, 54, 58]). For simplicity, in the following sections we will still use the abbreviation *FID*.

### 3.2 On the issue with default train-test splits in DML

To motivate the need for more comprehensive OOD evaluation protocols, we look at the split difficulty as measured by FID of typically used train-test splits and compare to i.i.d. sampling of training and test sets from the same benchmark. Empirical results in Tab. 1 show that commonly utilized DML

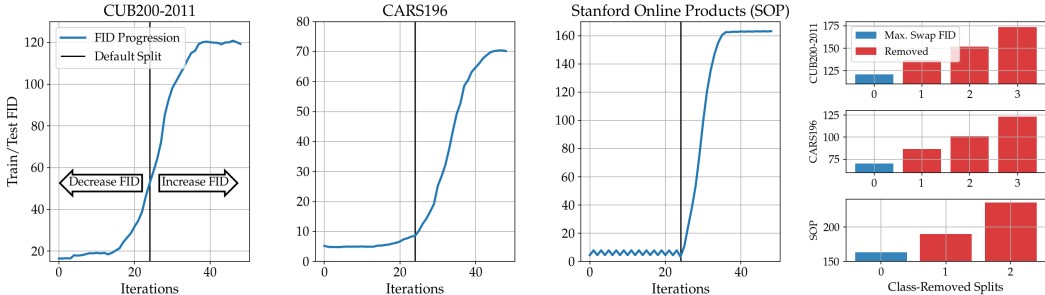

Figure 1: *FID progression with iterative class swapping and removal for train-test split generation.* (Col. 1-3) FID per swapping iteration $t$ on all benchmarks. (Rightmost) FID of data splits obtained by additional $k$ iterations of removing classes. The blue bar denotes the maximal FID after swapping.

train-test splits are very close to in-distribution learning problems when compared to more out-of-distribution splits in CARS196 and SOP (see Fig. 1). This indicates that semantic differences due to disjoint train and test classes, do not necessarily relate to actual significant distribution shifts between the train and test set. This also explains the consistently lower zero-shot retrieval performance on CUB200-2011 as compared to both CARS196 and SOP in literature [71, 70, 24, 54, 42, 38], despite SOP containing significantly more classes with fewer examples per class. In addition to the previously discussed issues of DML evaluation protocols, this further questions conclusions drawn from these protocols about the OOD generalization of representations $\phi$.

## 3.3 Creating train-test splits of increasing difficulty

Let $\mathcal{X}_{\text{train}}$ and $\mathcal{X}_{\text{test}}$ denote the original train and test set of a given benchmark dataset $\mathcal{D} = \mathcal{X}_{\text{train}} \cup \mathcal{X}_{\text{test}}$. To generate train-test splits of increasing difficulty while retaining the available data $\mathcal{D}$ and maintaining balance of their sizes, we exchange samples between them. To ensure and emphasize semantic consistency and unbiased data distributions with respect to image context unrelated to the target object categories, we swap entire classes instead of individual samples. Measuring distribution similarity based on FID, the goal is then to identify classes $C_{\text{train}} \subset \mathcal{X}_{\text{train}}$ and $C_{\text{test}} \subset \mathcal{X}_{\text{test}}$ whose exchange yields higher FID $d(\mathcal{X}_{\text{train}}, \mathcal{X}_{\text{test}})$. To this end, similar to other works [33, 51, 38], we find resorting to an *unimodal approximation* of the intraclass distributions sufficient and approximate FID by only considering the class means and neglect the covariance in Eq. 1. We select $C_{\text{train}}$ and $C_{\text{test}}$ as

$$C_{\text{train}}^* = \underset{C_{\text{train}} \in \mathcal{X}_{\text{train}}}{\arg\max} \|\mu_{C_{\text{train}}} - \mu_{\mathcal{X}_{\text{train}}}\|_2 - \|\mu_{C_{\text{train}}} - \mu_{\mathcal{X}_{\text{test}}}\|_2 \tag{2}$$

$$C_{\text{test}}^* = \underset{C_{\text{test}} \in \mathcal{X}_{\text{test}}}{\arg\max} \|\mu_{C_{\text{test}}} - \mu_{\mathcal{X}_{\text{test}}}\|_2 - \|\mu_{C_{\text{test}}} - \mu_{\mathcal{X}_{\text{train}}}\|_2 \tag{3}$$

where we measure distance to mean class-representations $\mu_{\mathcal{X}_C}$. By iteratively exchanging classes between data splits, i.e. $\mathcal{X}_{\text{train}}^{t+1} = (\mathcal{X}_{\text{train}}^t \setminus C_{\text{train}}^*) \cup C_{\text{test}}^*$ and vice versa, we obtain a more difficult train-test split $(\mathcal{X}_{\text{train}}^{t+1}, \mathcal{X}_{\text{test}}^{t+1})$ at iteration step $t$. Hence, we obtain a sequence of train-test splits $\boldsymbol{\mathcal{X}}_{\mathcal{D}} = ((\mathcal{X}_{\text{train}}^0, \mathcal{X}_{\text{test}}^0), \ldots, (\mathcal{X}_{\text{train}}^t, \mathcal{X}_{\text{test}}^t), \ldots, (\mathcal{X}_{\text{train}}^T, \mathcal{X}_{\text{test}}^T))$, with $\mathcal{X}_{\text{train}}^0 \triangleq \mathcal{X}_{\text{train}}$ and $\mathcal{X}_{\text{test}}^0 \triangleq \mathcal{X}_{\text{test}}$. Fig. 1 (columns 1-3) indeed shows that our FID approximation yields data splits with gradually increasing approximate FID scores with each swap until the scores cannot be further increased by swapping classes.

UMAP visualizations in the supplementary verify that the increase corresponds to larger OOD shifts. For CUB200-2011 and CARS196, we swap two classes per iteration, while for Stanford Online Products we swap 1000 classes due to a significantly higher class count. Moreover, to cover the overall spectrum of distribution shifts and ensure comparability between benchmarks we also reverse the iteration procedure on CUB200-2011 to generate splits minimizing the approximate FID while still maintaining disjunct train and test classes.

To further increase $d(\mathcal{X}_{\text{train}}^T, \mathcal{X}_{\text{test}}^T)$ beyond convergence (see Fig. 1) of the swapping procedure, we subsequently also identify and remove classes from both $\mathcal{X}_{\text{train}}^T$ and $\mathcal{X}_{\text{test}}^T$. More specifically, we remove classes $C_{\text{train}}$ from $\mathcal{X}_{\text{train}}^T$ that are closest to the mean of $\mathcal{X}_{\text{test}}^T$ and vice versa. For $k$ steps, we successively repeat class removal as long as $50\%$ of the original data is still maintained in these

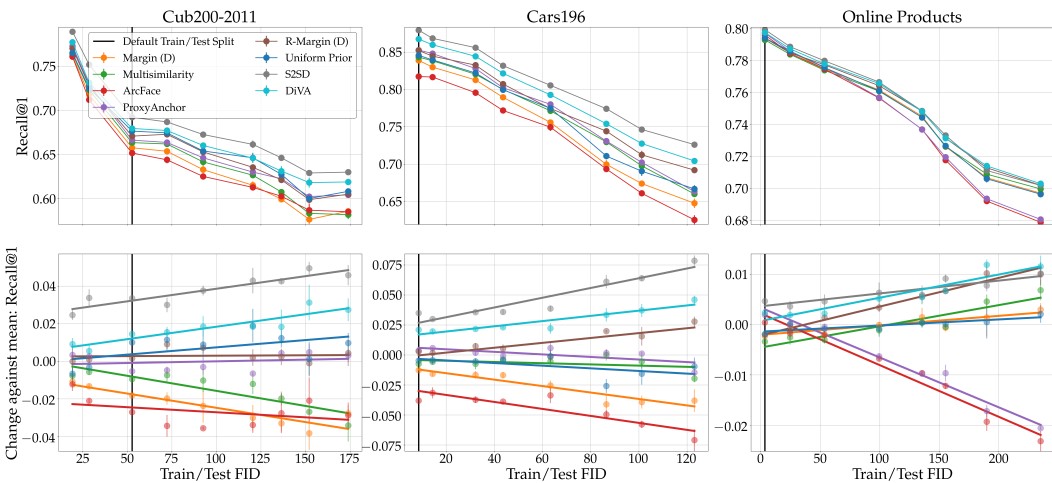

Figure 2: *Zero-Shot Generalization performance under varying distribution shifts.* (top row) Absolute Recall@1 performance for each increasingly more difficult train-test split in the *goodDML* benchmark (cf. Sec. 3) on CUB200-2011, CARS196 and SOP. We report mean Recall@1 performances and standard deviations over 5 runs. For results based on mAP@1000 see supplementary. (bottom) Differences of performances against the mean over all methods for each train-test split.

additional train-test splits. Fig. 1 (rightmost) shows how splits generated through class removal progressively increase the approximate FID beyond what was achieved only by swapping. To analyze if the generated data splits are not inherently biased to the used backbone network for FID computation, we also repeat this procedure based on representations from different architectures, pretraining methods and datasets in the supplementary. Note, that comparison of absolute FID values between datasets may not be meaningful and we are mainly interested in distribution shifts within a given dataset distribution. Overall, using class swapping and removal we select splits that cover the broadest FID range possible, while still maintaining sufficient data. Hence, our splits are significantly harder and more diverse than the default splits.

## 4   Assessing the State of Generalization in Deep Metric Learning

This section assesses the state of zero-shot generalization in DML via a large experimental study of representative DML methods on our *ooDML* benchmark, offering a much more complete and thorough perspective on zero-shot generalization in DML as compared to previous DML studies [13, 54, 42, 39].

For our experiments we use the three most widely used benchmarks in DML, CUB200-2011[68], CARS196[30] and Stanford Online Products[43]. For a complete list of implementation and training details see the supplementary if not explicitly stated in the respective sections. Moreover, to measure generalization performance, we resort to the most widely used metric for image retrieval in DML, Recall@k [25]. Additionally, we also evaluate results over mean average precision (mAP@1000) [54, 42], but provide respective tables and visualizations in the supplementary when the interpretation of results is not impacted.

The exact training and test splits ultimately utilized throughout this work are selected based on Fig. 1 to ensure approximately uniform coverage of the spectrum of distribution shifts within intervals ranging from the lowest (near i.i.d. splits) to the highest generated shift achieved with class removal. For experiments on CARS196 and Stanford Online Products, eight total splits were investigated, included the original default benchmark split. For CUB200-2011, we select nine splits to also account for benchmark additions with reduced distributional shifts. The exact FID ranges are provided in the supplementary. Training on CARS196 and CUB200-2011 was done for a maximum of 200 epochs following standard training protocols utilized in [54], while 150 epochs were used for the much larger SOP dataset. Additional training details if not directly stated in the respective sections can be found in the supplementary.

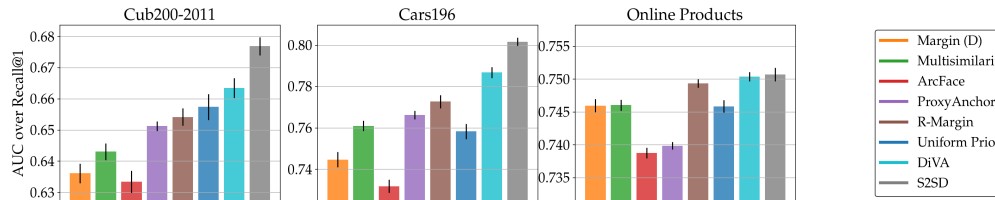

Figure 3: *Comparison of DML methods via AGS based on Recall@1 across benchmarks.* To compute AGS (cf. Sec. 4.1), we aggregate the performances from Fig. 2 across all train-test distribution shifts of our proposed *ooDML* benchmark using the Area-Under-the-Curve metric.

## 4.1 Zero-shot generalization under varying distribution shifts

Many different concepts have been proposed in DML to learn embedding functions $\phi$ that generalize from the training distribution to differently distributed test data. To analyze the zero-shot transfer capabilities of DML models, we consider representative approaches making use of the following concepts: *(i)* surrogate ranking tasks and tuple mining heuristics (Margin loss with distance-based sampling [71] and Multisimilarity loss [70]), *(ii)* geometric constraints or class proxies (ArcFace [8], ProxyAnchor [26]), *(iii)* learning of semantically diverse features (R-Margin [54]) and self-supervised training (DiVA [38]), adversarial regularization (Uniform Prior [59]) and *(iv)* knowledge self-distillation (S2SD [53]).

Fig. 2 (top) analyzes these methods for their generalization to distribution shifts the varying degrees represented in *ooDML*. The top row shows absolute zero-shot retrieval performance measured on Recall@1 (results for mAP@1000 can be found in the supplementary) with respect to the FID between train and test sets. Additionally, Fig. 2 (bottom) examines the relative differences of performance to the performance mean over all methods for each train-test split. Based on these experiments, we make the following main observations:

*(i)* **Performance deteriorates monotonically with the distribution shifts.** Independent of dataset, approach or evaluation metric, performance drops steadily as the distribution shift increases.

*(ii)* **Relative performance differences are affected by train-test split difficulty.** We see that the overall ranking between approaches oftentimes remains stable on the CARS196 and CUB200-2011 datasets. However, we also see that particularly on a large-scale dataset (SOP), established proxy-based approaches ArcFace [8] (which incorporates additional geometric constraints) and ProxyAnchor [26] are surprisingly susceptible to more difficult distribution shifts. Both methods perform poorly compared to the more consistent general trend of the other approaches. Hence, conclusions on the generality of methods solely based on the default benchmarks need to be handled with care, as at least for SOP, performance comparisons reported on single (e.g. the standard) data splits *do not* translate to more general train-test scenarios.

*(iii)* **Conceptual differences matter at larger distribution shifts** While the ranking between most methods is largely consistent on CUB200-2011 and CARS196, their differences in performance becomes more prominent with increasing distribution shifts. The relative changes (deviation from the mean of all methods at the stage) depicted in Fig. 2 (bottom) clearly indicates that particular methods based on machine learning techniques such as self-supervision, feature diversity (DiVA, R-Margin) and self-distillation (S2SD) are among the best at generalizing in DML on more challenging splits while retaining strong performance on more i.i.d. splits as well.

While directly stating performance in dependence to the individual distribution shifts offers a detailed overview, the overall comparison of approaches is typically based on single benchmark scores. To further provide a single metric of comparison, we utilize the well-known Area-under-Curve (AUC) score to condense performance (either based on Recall@1 or mAP@1000) over all available distribution shifts into a single aggregated score indicating general zero-shot capabilities. This *Aggregated Generalization Score (AGS)* is computed based on the normalized FID scores of our splits to the interval $[0, 1]$. As Recall@k or mAP@k scores are naturally bounded to $[0, 1]$, AGS is similarly bound to $[0, 1]$ with higher being the better model. Our corresponding results are visualized in Fig. 3. Indeed, we see that AGS reflects our observations from Fig. 2, with self-supervision (DiVA)

and self-distillation (S2SD) generally performing best when facing unknown train-test shifts. Exact scores are provided in the supplementary.

## 4.2 Consistency of structural representation properties on *ooDML*

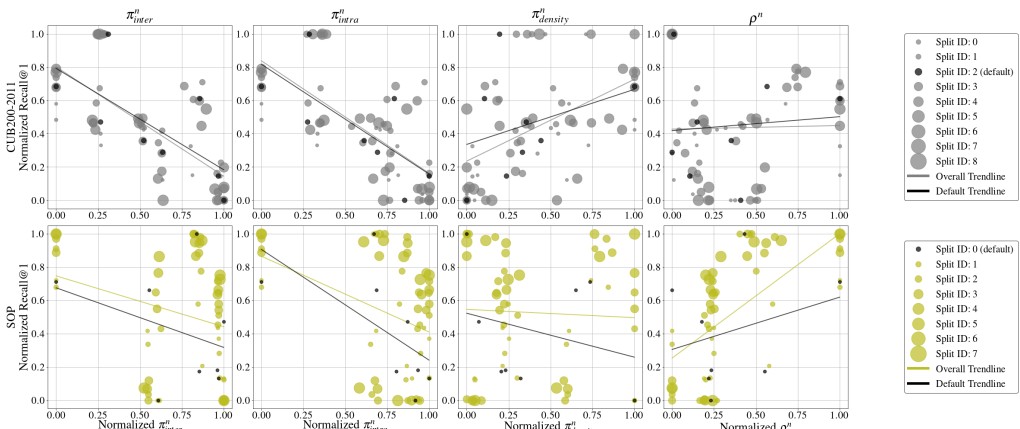

Figure 4: *Generalization metrics computed on ooDML benchmark for CUB200-2011 and SOP.* Each column plots one of the (normalized) measured structural representation property (cf. 4.2) over the corresponding Recall@1 performance for all examined DML methods and distribution shifts. Trendlines are computed as least squares fit over all datapoints (overall), respectively only those corresponding to default splits (default).

Roth et al. [54] attempts to identify potential drivers of generalization in DML by measuring the following structural properties of a representation $\phi$: *(i)* the mean distance $\pi_{\text{inter}}$ between the centers of the embedded samples of each class, *(ii)* the mean distance $\pi_{\text{intra}}$ between the embedded samples within a class $\pi_{\text{intra}}$, *(iii)* the 'embedding space density' measured as the ratio $\pi_{\text{ratio}} = \frac{\pi_{\text{intra}}}{\pi_{\text{inter}}}$ and *(iv)* 'spectral decay' $\rho(\Phi)$ measuring the degree of uniformity of singular values obtained by singular value decomposition on the training sample representations, which indicates the number of significant directions of variance. For a more detailed description, we refer to [54]. These metrics indeed are empirically shown to exhibit a certain correlation to generalization performance on the default benchmark splits. In contrast, we are now interested if these observations hold when measuring generalization performance on the *ooDML* train-test splits of varying difficulty.

We visualize our results in Fig. 4 for CUB200-2011 and SOP, with CARS196 provided in the supplementary. For better visualization we normalize all measured values obtained for both metrics *(i)-(iv)* and the recall performances (Recall@1) to the interval $[0, 1]$ for each train-test split. Thus, the relation between structural properties and generalization performance becomes comparable across all train-test splits, allowing us to examine if superior generalization is still correlated to the structural properties of the learned representation $\phi$, i.e. if the correlation is independent of the underlying distribution shifts. For a perfectly descriptive metric, one should expect to see heavy correlation between normalized metric and normalized generalization performance jointly across shifts. Unfortunately, our results show only a small indication of any structural metric being consistently correlated with generalization performance over varying distribution shifts. This is also observed when evaluating only against basic, purely discriminative DML objectives as was done in [54] for the default split, as well as when incorporating methods that extend and change the base DML training setup (such as DiVA [38] or adversarial regularization [59]).

This not only demonstrates that experimental conclusions derived from the analysis of only single benchmark split may not hold for overall zero-shot generalization, but also that future research should consider more general learning problems and difficulty to better understand the conceptual impact various regulatory approaches. To this end, our benchmark protocol offers more comprehensive experimental ground for future studies to find potential drivers of zero-shot generalization in DML.

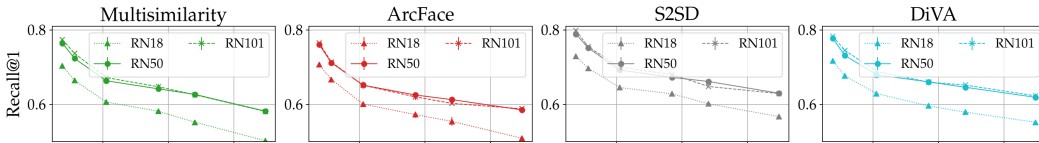

Figure 5: *Generalization performance for different backbone architectures for varying distribution shifts on CUB200-2011.* We show absolute Recall@1 performances averaged over 5 runs for each train-test split. Other datasets show similar results and are provided in the supplementary.

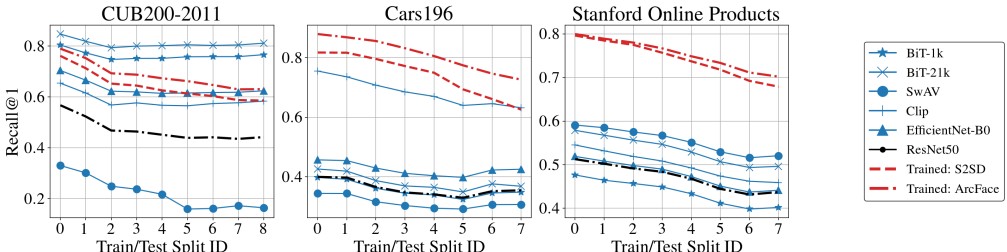

Figure 6: *Comparison of DML to various non-adapted generic representations pretrained on large amounts unlabelled data and state-of-the-art architectures.* For DML, we show best and worst DML objectives based on results in Fig. 2. Performance of generic representations are heavily dependent on target dataset, architecture, amount of training data and learning objective.

## 4.3 Network capacity and pretrained representations

A common way to improve generalization, as also highlighted in [54] and [42], is to select a stronger backbone architecture for feature extraction. In this section, we look at how changes in network capacity can influence OOD generalization across distribution shifts. Moreover, we also analyze the zero-shot performance of a diverse set of state-of-the-art pretraining approaches.

**Influence of network capacity.** In Fig. 5, we compare different members of the ResNet architecture family [20] with increasing capacity, each of which achieve increasingly higher performance on i.i.d. test benchmarks such as ImageNet [7], going from a small ResNet18 (R18) over ResNet50 (R50) to ResNet101 (R101) variants. As can be seen, while larger network capacity helps to some extent, we observe that performance actually saturates in zero-shot transfer settings, regardless of the DML approach and dataset (in particular also the large scale SOP dataset). Interestingly, we also observe that the performance drops with increasing distribution shifts are consistent across network capacity, suggesting that zero-shot generalization is less driven by network capacity but rather conceptual choices of the learning formulation (compare Fig. 2).

**Generic representations versus Deep Metric Learning.** Recently, self-supervised representation learning has taken great leaps with ever stronger models trained on huge amounts of image [29, 47] and language data [9, 35, 2]. These approaches are designed to learn expressive, well-transferring features and methods like CLIP [47] even prove surprisingly useful for zero-shot classification. We now evaluate and compare such representations against state-of-the-art DML models to understand if generic representations that are readily available nowadays actually pose an alternative to explicit application of DML. We select state-of-the-art self-supervision models *SwAV* [3] (ResNet50 backbone), CLIP [47] trained via natural language supervision on a large dataset of 400 million image and sentence pairs (VisionTransformer [10] backbone), BiT(-M) [29], which trains a ResNet50-V2 [29] on both the standard ImageNet [7] (1 million training samples) and the ImageNet-21k dataset [7, 50] with 14 million training samples and over 21 thousand classes, an EfficientNet-B0 [63] trained on ImageNet, and a standard baseline ResNet50 network trained on ImageNet. Note, that none of these representations has been additionally adapted to the benchmark sets, in contrast to the DML approaches which have been trained on the respective train splits.

The results presented in Fig. 6 show large performance differences of the pretrained representations, which are largely dependent on the test dataset. While BiT outperforms the DML state-of-the-art on CUB200-2011 without any finetuning, it significantly trails behind the DML models on the other

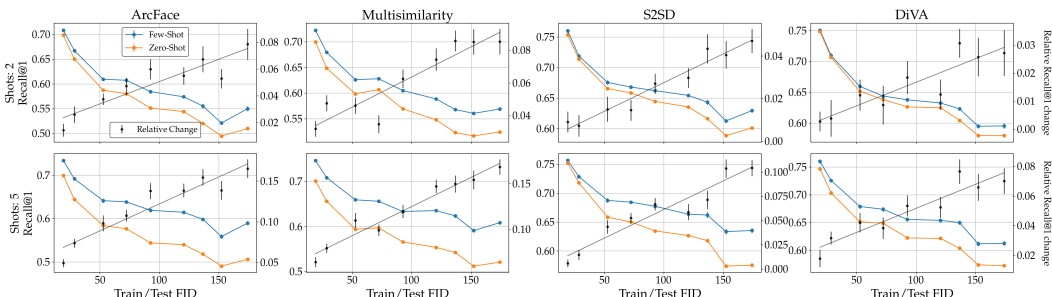

Figure 7: *Few-Shot adaptation of DML representations on CUB200-2011.* Columns show average Recall@1 performance over 10 episodes of 2- and 5-shot adaption for various DML approaches (fewshot and zeroshot), highlighting a substantial benefit of few-shot adaptation for *a priori* unknown distribution shifts (see black line highlighting relative improvements).

two datasets. On CARS196, only CLIP comes close to the DML approaches when the distribution shift is sufficiently large. Finally, on SOP, none of these models comes even close to the adapted DML methods. This shows how although representations learned by extensive pretraining can offer strong zero-shot generalization, their performance heavily depends on the target dataset and specific model. Furthermore, the generalization abilities notably depend on the size of the pretraining dataset (compare e.g. BiT-1k vs BiT-21k or CLIP), which is significantly larger than the number of training images seen by the DML methods. We see that only actual training on these datasets provides sufficiently reliable performance.

### 4.4 Few-shot adaption boosts generalization performance in DML

Since distribution shifts can be arbitrarily large, the zero-shot transfer of $\phi$ can be ill-posed. Features learned on a training set $\mathcal{X}_{\text{train}}$ will not meaningfully transfer to test samples $\mathcal{X}_{\text{test}}$ once they are sufficiently far from $\mathcal{X}_{\text{train}}$, as also already indicated by Fig. 2. As a remedy *few-shot learning* [60, 67, 14, 48, 32, 6, 66] assumes few samples of the test distribution to to be available during training, i.e. adjusting a previously learned representation. While these approaches are typically explicitly trained for fast adaption to novel classes, we are now interested if similar adaptation of DML representations $\phi$ helps to bridge increasingly large distribution shifts.

To investigate this hypothesis, we follow the evaluation protocol of few-shot learning and use $k$ representatives (also referred to as *shots*) of each class from a test set $\mathcal{X}_{\text{test}}$ as a support set for finetuning the penultimate embedding network layer. The remaining test samples then represent the new test set to evaluate retrieval performance, also referred to as query set. For evaluation we perform 10 episodes, i.e. we repeat and average the adaptation of $\phi$ over 10 different, randomly sampled support and corresponding query sets. Independent of the DML model used for learning the original representation $\phi$ on $\mathcal{X}_{\text{train}}$, adaptation to the support data is conducted using the Marginloss [71] objective with distance-based sampling [71] due to its faster convergence. This also ensures fair comparisons to the adaptation benefit to $\phi$ and also allows to adapt complex approaches like self-supervision (DiVA [38]) to the small number of samples in the support sets.

Fig. 7 shows 2 and 5 shot results on CUB200-2011, with CARS196 available in the supplementary. SOP is not considered since each class is already composed of small number of samples. As we see, even very limited in-domain data can significantly improve generalization performance, with the benefit becoming stronger for larger distribution shifts. Moreover, we observe that weaker approaches like ArcFace [8] seem to benefit more than state-of-the-art methods like S2SD [53] or DiVA [38]. We presume this to be caused by their underlying concepts already encouraging learning of more robust and general features. To conclude, few-shot learning provides a substantial and reliable benefit when facing OOD learning settings where the shift is not known *a priori*.

## 5 Conclusion

In this work we analyzed zero-shot transfer of image representations learned by Deep Metric Learning (DML) models. We proposed a systematic construction of train-test data splits of increasing

difficulty, as opposed to standard evaluation protocols that test out-of-distribution generalization only on single data splits of fixed difficulty. Based on this, we presented the novel benchmark *ooDML* and thoroughly assessed current DML methods. Our study reveals the following main findings:

**Standard evaluation protocols are insufficient to probe general out-of-distribution transfer:** Prevailing train-test splits in DML are often close to i.i.d. evaluation settings. Hence, they only provide limited insights into the impact of train-test distribution shift on generalization performance. Our benchmark *ooDML* alleviates this issue by evaluating a large, controllable and measurable spectrum of problem difficulty to facilitate future research.

**Larger distribution shifts show the impact of conceptual differences in DML approaches:** Our study reveals that generalization performance degrades consistently with increasing problem difficulty for all DML methods. However, certain concepts underlying the approaches are shown to be more robust to shifts than others, such as semantic feature diversity and knowledge-distillation.

**Generic, self-supervised representations without finetuning can surpass dedicated data adaptation:** When facing large distribution shifts, representations learned only by self-supervision on large amounts of of unlabelled data are competitive to explicit DML training without any finetuning. However, their performance is heavily dependent on the data distribution and the models themselves.

**Few-shot adaptation consistently improves out-of-distribution generalization in DML:** Even very few examples from a target data distribution effectively help to adapt DML representations. The benefit becomes even more prominent with increasing train-test distribution shifts, and encourages further research into few-shot adaptation in DML.

**Funding transparency statement**    This research has been funded by the German Federal Ministry for Economic Affairs and Energy within the project "KI-Absicherung – Safe AI for automated driving" and by the German Research Foundation (DFG) within projects 371923335 and 421703927. Moreover, it was funded in part by a CIFAR AI Chair at the Vector Institute, Microsoft Research, and an NSERC Discovery Grant. Resources used in preparing this research were provided, in part, by the Province of Ontario, the Government of Canada through CIFAR, and companies sponsoring the Vector Institute www.vectorinstitute.ai/#partners.

**Acknowledgements**    We thank the International Max Planck Research School for Intelligent Systems (IMPRS-IS) for supporting K.R; K.R. acknowledges his membership in the European Laboratory for Learning and Intelligent Systems (ELLIS) PhD program.

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
