# Supplementary: Characterizing Generalization under Out-Of-Distribution Shifts in Deep Metric Learning

## A  Analyzing the model bias for selecting train-test splits

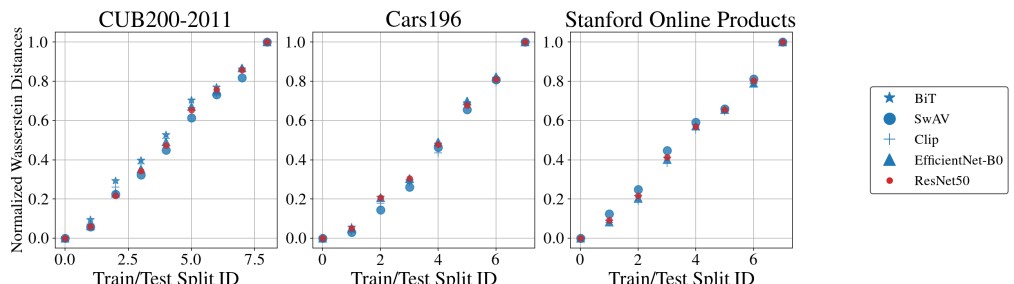

Figure 1: *Normalized FID progession for ooDML train-test splits using different training models and networks.* Values are normalized for comparability of FID progression, as FID scores are not upper bounded and as such, absolute values for different networks and pretraining methods differ.

To analyze the impact of the network architecture, pretraining method and training data, respectively the learned feature representations, on the construction of train-test splits and the entailed difficulties, we repeat our class swapping and removal procedure introduced in Section 3 in the main paper using different self-supervised models. Subsequently, we select train-test splits from the same iteration steps. Fig. 1 compares the progression of distribution shifts based on FID scores normalized to the $[0, 1]$ interval for valid comparison. We observe that across all pretrained models, the general FID progressions and sampled train-test splits exhibit very similar learning problem difficulties, indicating that our sampling procedure is robust to the choice of readily available, state-of-the art self-supervised pretrained models.

## B  Further Details regarding the Experimental Setup

**Datasets.**   In total, we utilized three widely used Deep Metric Learning benchmarks: (1) CUB200-2011 [15], which comprises a total of 11,788 images over 200 classes of birds, (2) CARS196 [5] containing 16,185 images of cars distributed over 196 classes and (3) Stanford Online Products (SOP) [10], which introduced 120,053 product images over 22,634 total classes. For CUB200-2011 and CARS196, default splits are simply generated by selecting the last half of the alphabetically sorted classes as test samples, whereas SOP provides a predefined split with 11318 training and 11316 test classes.

**Training details.**   For our implementation, we leveraged the PyTorch framework [11]. For training, in all cases, training images were randomly resized and cropped to $224 \times 224$, whereas for testing images were resized to $256 \times 256$ and center cropped to $224 \times 224$. Optimization was performed with Adam [4] with learning rate of $10^{-5}$ and weight decay of $3 \cdot 10^{-4}$. Batchsizes where chosen within the range of $[86, 112]$ depending on the size of the utilized backbone network. For default DML ResNet-architectures, we follow previous literature [18, 3, 14, 9] and freeze Batch-Normalization

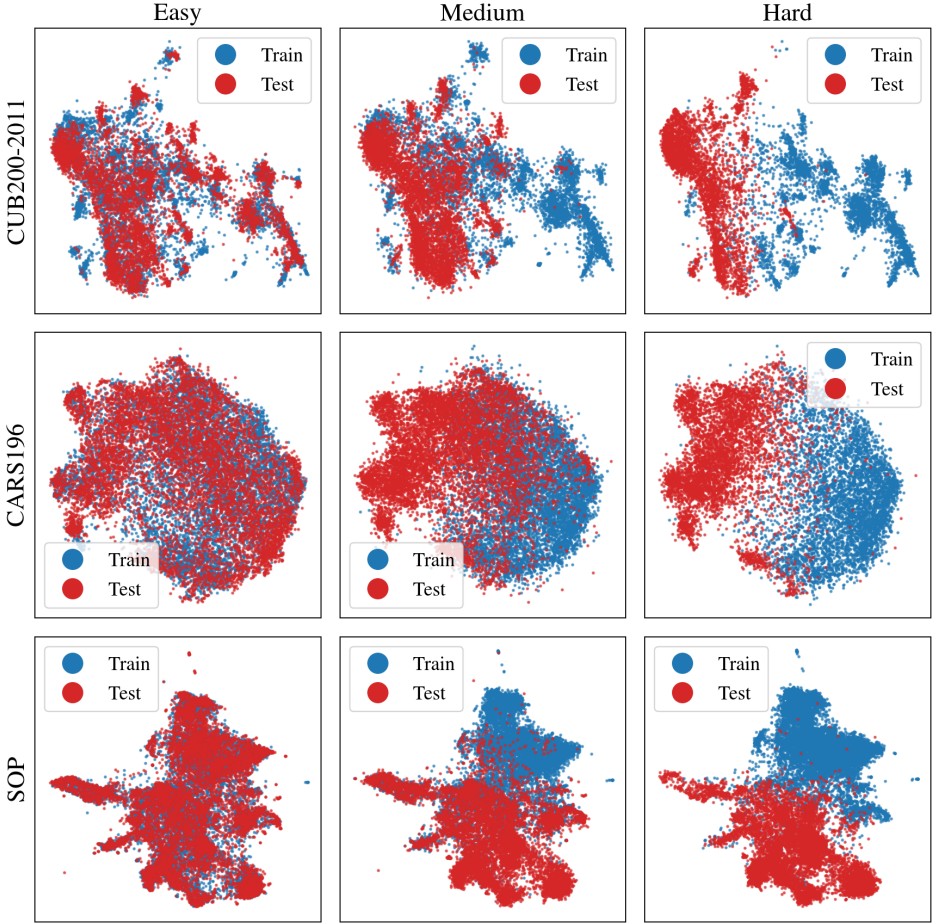

Figure 2: *UMAPs* for easy (split-id 1), medium (split-id 5) and hard (split-id 8/9) for all benchmarks using a ResNet50 backbone pretrained on ImageNet. As can be seen, our iterative class swapping (and removal) procedure (cf. Sec. 3.2 main paper) creates splits in which training and test distributions become increasingly disjoint. Note that while we shifted full classes for semantic consistency, each point corresponds to a single sample (for SOP, random subsampling to 20000 total points was performed).

layers during training. We consistently use an embedding dimensionality of 512 for comparability. For DiVA [8], S2SD [13] and ProxyAnchor [3], parameter choices were set to default values given in the original publications, with small grid searches done to allow for adaptation to backbone changes. For all other remaining objectives, parameter choices were adapted from [14], who provide a hyperparameter selection for best comparability of methods. All experiments were performed on GPU servers containing NVIDIA P100, T4 and Titan X, with results always averaged over multiple seeds - in the case of our objective study five random seeds were utilized, whereas for other ablation-type studies at least three seeds were utilized. These settings are used throughout our study. For the few-shot experiments, the same pipeline parameters were utilized with changes noted in the respective section.

Pretrained network weights for ResNet-architectures where taken from torchvision [6], EfficientNet and BiT weights from timm [17] and SwAV and CLIP pretrained weights from the respective official repositories ([1] and [12]).

**FID scores between *ooDML* data splits.** In Tab. 1 we show the measured FID scores between each train-test split of our *ooDML* for the CUB200-2011, CARS196 and SOP benchmarks, respectively.

Table 1: FID scores between train-test splits in our ooDML benchmark. For details on creating train-test splits constituting the *ooDML* benchmark, please see Sec. 3 in main paper.

| Dataset ↓ split-ID → | 1 | 2 | 3 | 4 | 5 | 6 | 7 | 8 | 9 |
|---|---|---|---|---|---|---|---|---|---|
| CUB200-2011 | 19.2 | 28.5 | 52.6 | 72.2 | 92.5 | 120.4 | 136.5 | 152.0 | 173.9 |
| CARS196 | 8.6 | 14.3 | 32.2 | 43.6 | 63.3 | 86.5 | 101.2 | 123.0 | - |
| SOP | 3.4 | 24.6 | 53.5 | 99.4 | 135.5 | 155.3 | 189.8 | 235.1 | - |

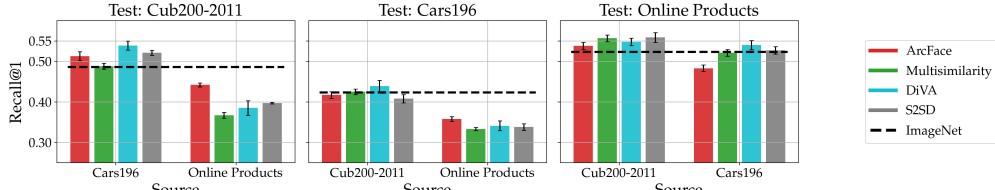

Figure 3: *Out-of-Domain Generalization.* Each plot showcases transfer performance from the training dataset (*source*) to a test dataset from a novel domain (*test*). The dashed line represents baseline performance achieved by ResNet50 pretrained on ImageNet.

**Qualitative introspection of *ooDML* train-test splits using UMAP.** Fig. 2 visualizes the distribution shift between train-test splits from our proposed *ooDML* benchmark using the UMAP [7] algorithm. For each dataset we show examples for an easy, medium and hard train-test split. Indeed, the distributional shift train to test data is increasing consistently, as indicated by our monotonically increasing FID progressions.

## C   On the limits of OOD generalization in Deep Metric Learning

To investigate how well representations $\phi$ learned by DML approaches transfer *across* benchmark datasets, we train our models on the default training dataset of one benchmark and evaluate them on the default test set of another. Tab. 2 first illustrates the FID scores for all pairwise combinations using the CUB200-2011, CARS196 and SOP datasets. We find all FID scores exceed the previously considered learning problems in our *ooDML* benchmark by far. However, the fact that FID scores are relatively close to another despite large semantic differences between datasets may indicate that FID based on our utilised FID estimator (Sec. 3 main paper) may have reached its limit as a distributional shift indicator, thus not being sufficiently sensitive. Fig. 3 summarizes the generalization performances for different DML approaches on this experimental setting. As can be seen, there are only a few cases where $\phi$ offers a benefit over the ResNet50 ImageNet baseline, indicating that generalization of DML approaches is primarily limited to shifts within a data domains. Beyond these limits, generic representations learned by self-supervised learning may offer better zero-shot generalization, as also discussed on Sec. 4.4.

## D   Additional Experimental Results

### D.1   Zero-shot generalization under varying distribution shifts

This section provides additional results for the experiments presented in Sec. 4 in the main paper. To this end, we provide the exact performance values used to visualize Fig. 2 in the main paper in Tab. 4-6. For the comparison based on the Aggregated Generalization Score (AGS) introduced in Sec. 4.2 in the main paper, Tab. 3 provides the empirical results both for AGS computed based on Recall1@1 and mAP@1000. For the latter, Fig. 4 summarizes AGS results using a bar plot similar to Fig. 3 in the main paper.

### D.2   Influence of network capacity

In Fig. 5 we present all results for our study on the influence of network capacity in Sec. 4.4 in the main paper, in particular also for the remaining datasets CARS196 and SOP. Additionally, we show

Table 2: FID scores between training and test sets across different datasets compared to the highest FID measured by our generated train-test splits.

| Direction (train→test) | CUB→CARS | CARS→CUB | CUB→SOP | SOP→CUB | CARS→SOP | SOP→CARS | Max. *ooDML* |
|---|---|---|---|---|---|---|---|
| **FID** | 349 | 359 | 359 | 370 | 386 | 376 | 155 (235) |

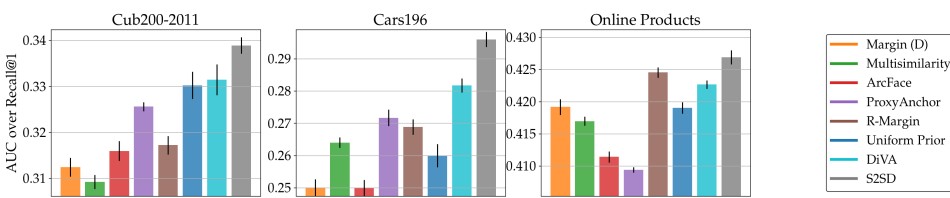

Figure 4: *Comparison of DML methods via AGS based on mAP@1000 across benchmarks.* To compute AGS (cf. Sec. 4.2 main paper), we aggregate the mAP@1000 performances in Tab. 4-6 across all train-test distribution shifts of our proposed *ooDML* benchmark using the Area-Under-the-Curve metric.

the differences in performances against the mean over all methods for each train-test split (Change against mean). As already discussed in Sec. **??** in the main paper, these experiments similarly show that network capacity has only limited impact on OOD generalization, with benefits saturating eventually.

### D.3 Measuring structural representation properties on ooDML

This section extends the results presented in Sec. 4.3 in the main paper. We show results for all datasets, i.e. CUB200-2011, CARS196 and SOP, for all metrics measuring structural representation properties discussed in Sec. 4.3 in the main paper. We analyze correlations of these metrics with generalization performance both based on Recall@1 (Fig. 6) and mAP@1000 (Fig. 7). As discussed in the main paper, independent of the underlying performance metric, none of the structural representation properties show consistent correlation with generalization performance across all datasets, suggesting further research into meaningful latent space properties that can be linked to zero-shot generalization independent of chosen objectives and shifts.

### D.4 Few-Shot DML

In Sec. 4.5 in the main paper, we analyzed few-shot adaption of DML representations to novel test distributions as a remedy to bridge their distribution shift to the training data. This section extends showcased results: Fig. 8 presents all our results on both CUB200-2011 (a+b) and CARS196 (c+d) dataset based on both Recall@1 and mAP@1000. The results on CARS196 verify the consistent improvement of leveraging very few examples for embedding space adaption over strict zero-shot transfer based on the original DML representation that we already observed for the CUB200-2011 dataset, which holds disproportionally well for larger distribution shifts. The corresponding data basis for Fig. 8 is presented in Tab. 7 for the CUB200-2011 dataset and in Tab. 8 for the CARS196 dataset.

Table 3: Results for Aggregated Generalization Score (AGS) (cf. Sec. 4.2 main paper) based on Recall@1 and mAP@1000 computed on the *ooDML* benchmark. We show results for various DML methods averaged over over multiple runs.

| Benchmark→ | CUB200-2011 | | CARS196 | | SOP | |
|---|---|---|---|---|---|---|
| **Approaches ↓ AUC →** | R@1 | mAP@1000 | R@1 | mAP@1000 | R@1 | mAP@1000 |
| Margin (D) [18] | $63.6 \pm 0.3$ | $31.2 \pm 0.2$ | $74.5 \pm 0.4$ | $25.0 \pm 0.3$ | $74.6 \pm 0.1$ | $41.9 \pm 0.1$ |
| Multisimilarity [16] | $64.3 \pm 0.3$ | $30.9 \pm 0.2$ | $76.1 \pm 0.2$ | $26.4 \pm 0.2$ | $74.6 \pm 0.1$ | $41.7 \pm 0.1$ |
| ArcFace [2] | $63.3 \pm 0.4$ | $31.6 \pm 0.2$ | $73.2 \pm 0.3$ | $25.0 \pm 0.3$ | $73.9 \pm 0.1$ | $41.1 \pm 0.1$ |
| ProxyAnchor [3] | $65.1 \pm 0.2$ | $32.6 \pm 0.1$ | $76.6 \pm 0.2$ | $27.2 \pm 0.3$ | $74.0 \pm 0.1$ | $40.9 \pm 0.1$ |
| R-Margin [14] | $65.4 \pm 0.3$ | $31.7 \pm 0.2$ | $77.3 \pm 0.3$ | $26.9 \pm 0.2$ | $74.9 \pm 0.1$ | $42.5 \pm 0.1$ |
| Uniform Prior | $65.7 \pm 0.5$ | $33.0 \pm 0.3$ | $75.8 \pm 0.4$ | $26.0 \pm 0.4$ | $74.6 \pm 0.1$ | $41.9 \pm 0.1$ |
| DiVA [8] | $66.4 \pm 0.3$ | $33.1 \pm 0.3$ | $78.6 \pm 0.3$ | $27.9 \pm 0.2$ | $75.0 \pm 0.1$ | $42.3 \pm 0.1$ |
| S2SD | $\mathbf{67.7} \pm 0.3$ | $\mathbf{33.9} \pm 0.2$ | $\mathbf{80.2} \pm 0.2$ | $\mathbf{29.6} \pm 0.2$ | $\mathbf{75.1} \pm 0.1$ | $\mathbf{42.7} \pm 0.1$ |

Table 4: DML generalization performance measured by Recall@1 and mAP@1000 on each train-test split of our *ooDML* benchmark for the CUB200-2011 dataset.

| Metric | Method↓ \| Split (FID)→ | 1 (19.2) | 2 (28.5) | 3 (52.6) | 4 (72.2) | 5 (92.5) | 6 (120.4) | 7 (136.5) | 8 (152.0) | 9 (173.9) |
|---|---|---|---|---|---|---|---|---|---|---|
| R@1 | Margin (D) | 76.20 ± 0.13 | 71.79 ± 0.09 | 65.78 ± 0.05 | 65.38 ± 0.22 | 63.30 ± 0.58 | 61.53 ± 0.43 | 59.95 ± 0.27 | 57.67 ± 0.53 | 58.59 ± 0.27 |
| | Multisimilarity | 76.44 ± 0.42 | 72.34 ± 0.11 | 66.36 ± 0.24 | 66.21 ± 0.26 | 64.18 ± 0.40 | 62.68 ± 0.34 | 60.77 ± 0.23 | 58.35 ± 0.06 | 58.20 ± 0.51 |
| | ArcFace | 76.10 ± 0.28 | 71.22 ± 0.08 | 65.19 ± 0.38 | 64.41 ± 0.39 | 62.53 ± 0.05 | 61.29 ± 0.27 | 60.28 ± 0.66 | 58.71 ± 0.74 | 58.53 ± 0.41 |
| | ProxyAnchor | 77.30 ± 0.14 | 72.95 ± 0.13 | 66.64 ± 0.09 | 66.39 ± 0.13 | 64.64 ± 0.26 | 63.03 ± 0.07 | 62.27 ± 0.08 | 60.25 ± 0.25 | 60.44 ± 0.21 |
| | R-Margin (D) | 77.04 ± 0.72 | 72.82 ± 0.18 | 67.10 ± 0.28 | 67.31 ± 0.17 | 65.31 ± 0.20 | 63.57 ± 0.28 | 62.14 ± 0.27 | 59.90 ± 0.39 | 60.52 ± 0.42 |
| | Uniform Prior | 76.53 ± 0.30 | 72.52 ± 0.17 | 67.67 ± 0.36 | 67.47 ± 0.39 | 65.42 ± 0.53 | 64.64 ± 0.56 | 62.76 ± 0.31 | 60.03 ± 0.55 | 60.84 ± 0.32 |
| | S2SD | 78.93 ± 0.20 | 75.20 ± 0.33 | 69.24 ± 0.51 | 68.70 ± 0.26 | 67.28 ± 0.17 | 66.16 ± 0.43 | 64.64 ± 0.08 | 62.93 ± 0.20 | 63.02 ± 0.32 |
| | DiVA | 77.74 ± 0.25 | 73.15 ± 0.26 | 67.97 ± 0.26 | 67.74 ± 0.19 | 66.04 ± 0.36 | 64.61 ± 0.12 | 63.14 ± 0.57 | 61.83 ± 0.57 | 61.91 ± 0.36 |
| mAP@1000 | Margin (D) | 44.76 ± 0.27 | 39.69 ± 0.20 | 34.31 ± 0.09 | 33.59 ± 0.13 | 30.64 ± 0.29 | 28.17 ± 0.16 | 26.72 ± 0.31 | 25.45 ± 0.18 | 26.18 ± 0.33 |
| | Multisimilarity | 44.21 ± 0.11 | 39.03 ± 0.07 | 33.79 ± 0.01 | 33.38 ± 0.26 | 30.75 ± 0.19 | 27.83 ± 0.15 | 26.58 ± 0.15 | 25.25 ± 0.22 | 25.29 ± 0.23 |
| | ArcFace | 45.45 ± 0.23 | 40.39 ± 0.37 | 34.39 ± 0.14 | 34.08 ± 0.06 | 31.06 ± 0.15 | 27.58 ± 0.15 | 27.31 ± 0.44 | 26.19 ± 0.28 | 26.90 ± 0.22 |
| | ProxyAnchor | 46.20 ± 0.11 | 40.99 ± 0.14 | 35.23 ± 0.03 | 34.89 ± 0.13 | 32.06 ± 0.04 | 28.88 ± 0.06 | 28.15 ± 0.18 | 27.31 ± 0.18 | 28.18 ± 0.02 |
| | R-Margin (D) | 44.76 ± 0.41 | 39.77 ± 0.15 | 34.28 ± 0.09 | 34.56 ± 0.29 | 31.46 ± 0.11 | 28.25 ± 0.06 | 27.24 ± 0.25 | 26.48 ± 0.37 | 26.64 ± 0.39 |
| | Uniform Prior | 46.00 ± 0.62 | 41.04 ± 0.40 | 36.07 ± 0.23 | 35.51 ± 0.33 | 32.57 ± 0.18 | 30.15 ± 0.33 | 28.64 ± 0.20 | 27.21 ± 0.23 | 27.73 ± 0.46 |
| | S2SD | 47.19 ± 0.08 | 42.27 ± 0.40 | 36.49 ± 0.02 | 36.22 ± 0.19 | 33.79 ± 0.10 | 30.69 ± 0.08 | 29.11 ± 0.09 | 28.51 ± 0.42 | 28.83 ± 0.30 |
| | DiVA | 46.74 ± 0.51 | 41.07 ± 0.71 | 35.80 ± 0.24 | 35.63 ± 0.24 | 32.63 ± 0.20 | 30.02 ± 0.25 | 28.33 ± 0.31 | 27.88 ± 0.49 | 28.86 ± 0.30 |

Table 5: DML generalization performance measured by Recall@1 and mAP@1000 on each train-test split of our *ooDML* benchmark for the CARS196 dataset.

| Metric | Method↓ \| Split (FID)→ | 1 (8.6) | 2 (14.3) | 3 (32.2) | 4 (43.6) | 5 (63.3) | 6 (86.5) | 7 (101.2) | 8 (123.0) |
|---|---|---|---|---|---|---|---|---|---|
| R@1 | Margin (D) | 83.89 ± 0.24 | 82.99 ± 0.15 | 81.27 ± 0.26 | 78.95 ± 0.20 | 75.59 ± 0.32 | 69.97 ± 0.61 | 67.41 ± 0.38 | 64.77 ± 0.63 |
| | Multisimilarity | 84.33 ± 0.21 | 83.84 ± 0.10 | 82.03 ± 0.38 | 80.01 ± 0.06 | 77.14 ± 0.22 | 72.97 ± 0.34 | 69.78 ± 0.37 | 66.01 ± 0.21 |
| | ArcFace | 81.73 ± 0.29 | 81.66 ± 0.39 | 79.57 ± 0.23 | 77.19 ± 0.06 | 74.95 ± 0.50 | 69.35 ± 0.26 | 66.10 ± 0.21 | 62.55 ± 0.62 |
| | ProxyAnchor | 85.27 ± 0.17 | 84.81 ± 0.05 | 82.80 ± 0.22 | 80.23 ± 0.19 | 78.00 ± 0.40 | 73.09 ± 0.21 | 70.24 ± 0.09 | 66.35 ± 0.17 |
| | R-Margin (D) | 85.21 ± 0.15 | 84.46 ± 0.34 | 83.26 ± 0.22 | 80.73 ± 0.28 | 77.45 ± 0.46 | 74.42 ± 0.07 | 71.25 ± 0.55 | 69.20 ± 0.28 |
| | Uniform Prior | 84.56 ± 0.16 | 83.96 ± 0.30 | 82.20 ± 0.15 | 79.96 ± 0.18 | 77.48 ± 0.37 | 71.78 ± 0.42 | 69.06 ± 0.37 | 66.69 ± 0.50 |
| | S2SD | 87.93 ± 0.07 | 86.84 ± 0.08 | 85.59 ± 0.10 | 83.18 ± 0.17 | 80.55 ± 0.16 | 77.42 ± 0.41 | 74.64 ± 0.16 | 72.62 ± 0.29 |
| | DiVA | 86.74 ± 0.08 | 85.98 ± 0.14 | 84.43 ± 0.11 | 82.16 ± 0.08 | 79.28 ± 0.35 | 75.41 ± 0.42 | 72.76 ± 0.38 | 70.43 ± 0.22 |
| mAP@1000 | Margin (D) | 33.58 ± 0.26 | 33.23 ± 0.01 | 30.33 ± 0.28 | 28.50 ± 0.28 | 25.99 ± 0.24 | 21.43 ± 0.17 | 18.76 ± 0.53 | 16.56 ± 0.22 |
| | Multisimilarity | 34.01 ± 0.29 | 34.37 ± 0.20 | 31.39 ± 0.28 | 29.82 ± 0.27 | 28.09 ± 0.16 | 22.72 ± 0.01 | 20.39 ± 0.09 | 17.38 ± 0.15 |
| | ArcFace | 33.93 ± 0.20 | 34.19 ± 0.22 | 30.85 ± 0.07 | 28.51 ± 0.28 | 26.71 ± 0.50 | 20.67 ± 0.24 | 18.20 ± 0.10 | 15.40 ± 0.29 |
| | ProxyAnchor | 35.83 ± 0.17 | 36.22 ± 0.20 | 32.71 ± 0.19 | 31.07 ± 0.24 | 29.04 ± 0.58 | 23.08 ± 0.08 | 20.26 ± 0.26 | 17.16 ± 0.10 |
| | R-Margin (D) | 34.34 ± 0.24 | 34.69 ± 0.20 | 32.25 ± 0.28 | 30.63 ± 0.29 | 28.02 ± 0.16 | 23.12 ± 0.30 | 20.70 ± 0.34 | 18.68 ± 0.06 |
| | Uniform Prior | 34.03 ± 0.23 | 34.22 ± 0.38 | 31.05 ± 0.52 | 29.56 ± 0.38 | 26.99 ± 0.25 | 22.19 ± 0.29 | 20.06 ± 0.46 | 17.80 ± 0.28 |
| | S2SD | 37.41 ± 0.14 | 37.43 ± 0.18 | 34.48 ± 0.15 | 33.18 ± 0.24 | 30.93 ± 0.32 | 26.12 ± 0.15 | 23.56 ± 0.25 | 21.02 ± 0.36 |
| | DiVA | 36.60 ± 0.40 | 36.65 ± 0.09 | 32.96 ± 0.40 | 31.90 ± 0.08 | 29.40 ± 0.24 | 24.21 ± 0.25 | 22.10 ± 0.24 | 19.77 ± 0.16 |

Table 6: DML generalization performance measured by Recall@1 and mAP@1000 on each train-test split of our *ooDML* benchmark for the SOP dataset.

| Metric | Method↓ \| Split (FID)→ | 1 (3.4) | 2 (24.6) | 3 (53.5) | 4 (99.4) | 5 (135.5) | 6 (155.3) | 7 (189.8) | 8 (235.1) |
|---|---|---|---|---|---|---|---|---|---|
| R@1 | Margin (D) | 79.39 ± 0.04 | 78.58 ± 0.05 | 77.48 ± 0.05 | 76.18 ± 0.05 | 74.53 ± 0.14 | 72.62 ± 0.05 | 70.68 ± 0.27 | 69.71 ± 0.09 |
| | Multisimilarity | 79.31 ± 0.09 | 78.40 ± 0.11 | 77.40 ± 0.06 | 76.10 ± 0.10 | 74.45 ± 0.07 | 72.63 ± 0.01 | 70.90 ± 0.10 | 69.98 ± 0.14 |
| | ArcFace | 79.61 ± 0.07 | 78.46 ± 0.10 | 77.48 ± 0.12 | 75.67 ± 0.03 | 73.70 ± 0.04 | 71.78 ± 0.04 | 69.21 ± 0.13 | 67.89 ± 0.14 |
| | ProxyAnchor | 79.73 ± 0.07 | 78.60 ± 0.02 | 77.60 ± 0.05 | 75.70 ± 0.05 | 73.69 ± 0.05 | 71.96 ± 0.12 | 69.36 ± 0.09 | 68.07 ± 0.03 |
| | R-Margin (D) | 79.42 ± 0.01 | 78.50 ± 0.01 | 77.75 ± 0.05 | 76.44 ± 0.07 | 74.84 ± 0.03 | 73.15 ± 0.00 | 71.30 ± 0.14 | 70.21 ± 0.16 |
| | Unifor mPrior | 79.42 ± 0.02 | 78.61 ± 0.04 | 77.57 ± 0.05 | 76.12 ± 0.07 | 74.45 ± 0.08 | 72.68 ± 0.06 | 70.61 ± 0.23 | 69.65 ± 0.12 |
| | S2SD | 79.95 ± 0.06 | 78.88 ± 0.09 | 78.00 ± 0.11 | 76.66 ± 0.15 | 74.86 ± 0.14 | 73.33 ± 0.06 | 71.13 ± 0.08 | 70.19 ± 0.09 |
| | DiVA | 79.76 ± 0.08 | 78.75 ± 0.05 | 77.81 ± 0.06 | 76.58 ± 0.05 | 74.83 ± 0.09 | 73.15 ± 0.03 | 71.42 ± 0.07 | 70.30 ± 0.15 |
| mAP@1000 | Margin (D) | 47.47 ± 0.03 | 46.21 ± 0.10 | 45.16 ± 0.09 | 43.39 ± 0.04 | 41.79 ± 0.18 | 40.13 ± 0.03 | 37.64 ± 0.31 | 36.43 ± 0.09 |
| | Multisimilarity | 47.23 ± 0.07 | 45.85 ± 0.09 | 44.81 ± 0.03 | 43.17 ± 0.03 | 41.46 ± 0.07 | 39.67 ± 0.10 | 37.56 ± 0.07 | 36.76 ± 0.13 |
| | ArcFace | 47.76 ± 0.05 | 46.22 ± 0.10 | 45.09 ± 0.05 | 42.85 ± 0.07 | 40.86 ± 0.10 | 39.09 ± 0.09 | 36.01 ± 0.11 | 34.68 ± 0.13 |
| | ProxyAnchor | 47.57 ± 0.04 | 45.87 ± 0.02 | 44.89 ± 0.03 | 42.52 ± 0.09 | 40.67 ± 0.01 | 38.93 ± 0.06 | 35.90 ± 0.05 | 34.65 ± 0.00 |
| | R-Margin (D) | 47.94 ± 0.04 | 46.57 ± 0.01 | 45.59 ± 0.05 | 43.87 ± 0.00 | 42.26 ± 0.11 | 40.68 ± 0.07 | 38.44 ± 0.19 | 37.08 ± 0.14 |
| | Uniform Prior | 47.49 ± 0.02 | 46.28 ± 0.07 | 45.21 ± 0.05 | 43.39 ± 0.05 | 41.67 ± 0.12 | 40.07 ± 0.01 | 37.62 ± 0.21 | 36.37 ± 0.10 |
| | S2SD | 48.25 ± 0.04 | 47.08 ± 0.18 | 46.09 ± 0.09 | 44.27 ± 0.07 | 42.34 ± 0.15 | 40.87 ± 0.11 | 38.39 ± 0.10 | 36.98 ± 0.11 |
| | DiVA | 48.08 ± 0.06 | 46.57 ± 0.05 | 45.50 ± 0.05 | 43.77 ± 0.04 | 41.92 ± 0.06 | 40.44 ± 0.10 | 38.10 ± 0.10 | 36.73 ± 0.07 |

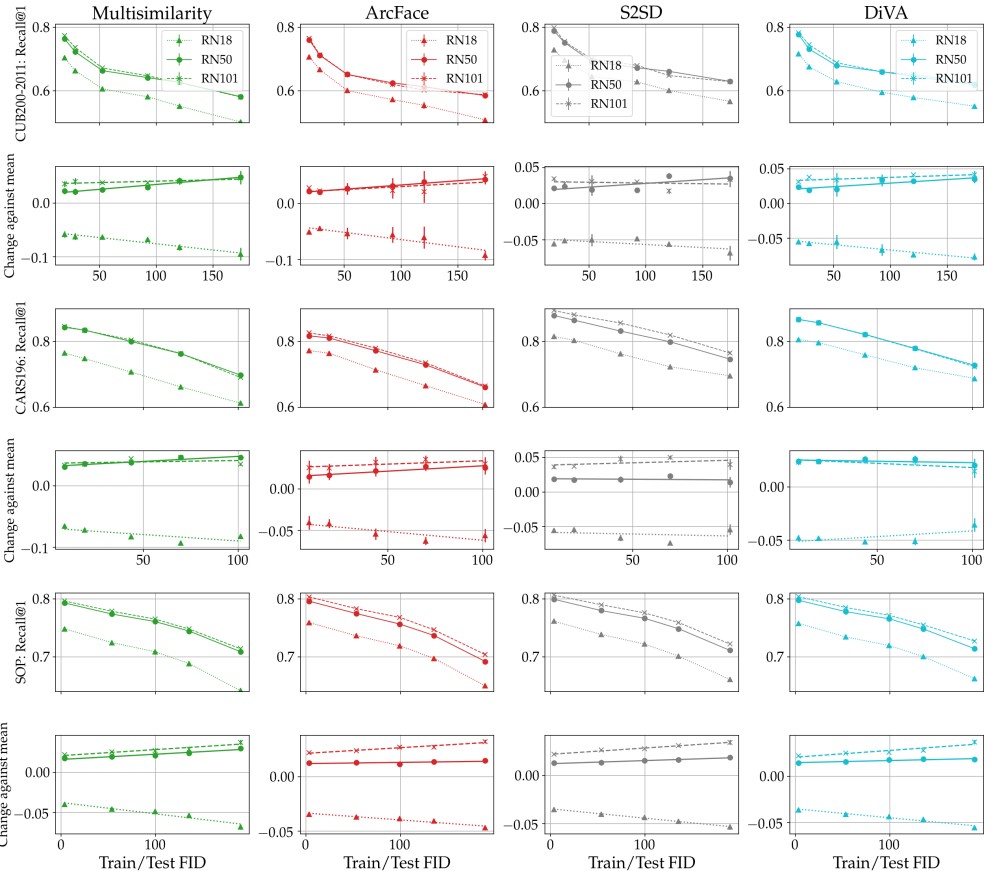

Figure 5: *Generalization performance for different backbone architectures for varying distribution shifts on full ooDML benchmark (CUB200-2011, CARS196, SOP).* To reduce computational load, we only utilised two thirds of the studied splits. Overall, we show absolute Recall@1 performances averaged over 5 runs for each train-test split.

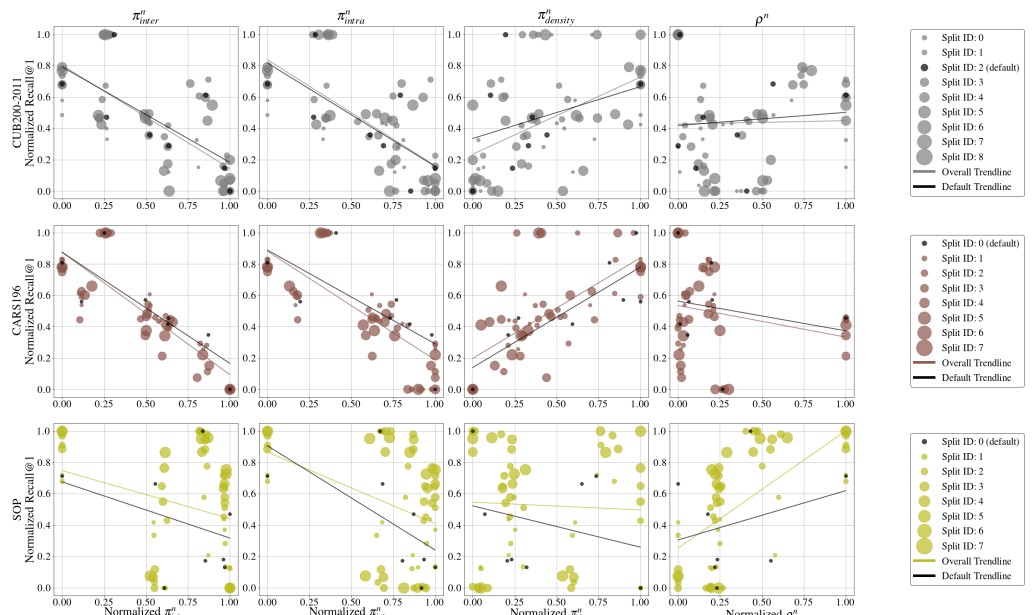

Figure 6: *Generalization metrics computed on ooDML benchmark for all datasets* measured against Recall@1. Each column plots one of the (normalized) measured structural representation property (cf. Sec. 4.3 main paper) over the corresponding Recall@1 performance for all examined DML methods and distribution shifts. Trendlines are computed as least squares fit over all datapoints (overall), respectively only those corresponding to default splits (default).

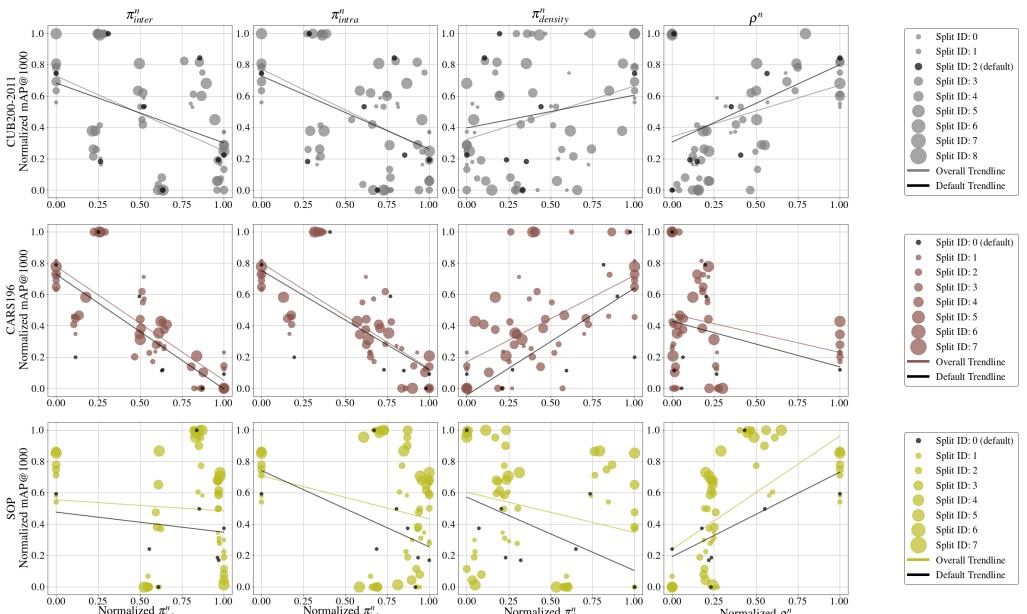

Figure 7: *Generalization metrics computed on ooDML benchmark for all datasets* measured against mAP@1000. Each column plots one of the (normalized) measured structural representation property (cf. Sec. 4.3 main paper) over the corresponding mAP@1000 performance for all examined DML methods and distribution shifts. Trendlines are computed as least squares fit over all datapoints (overall), respectively only those corresponding to default splits (default).

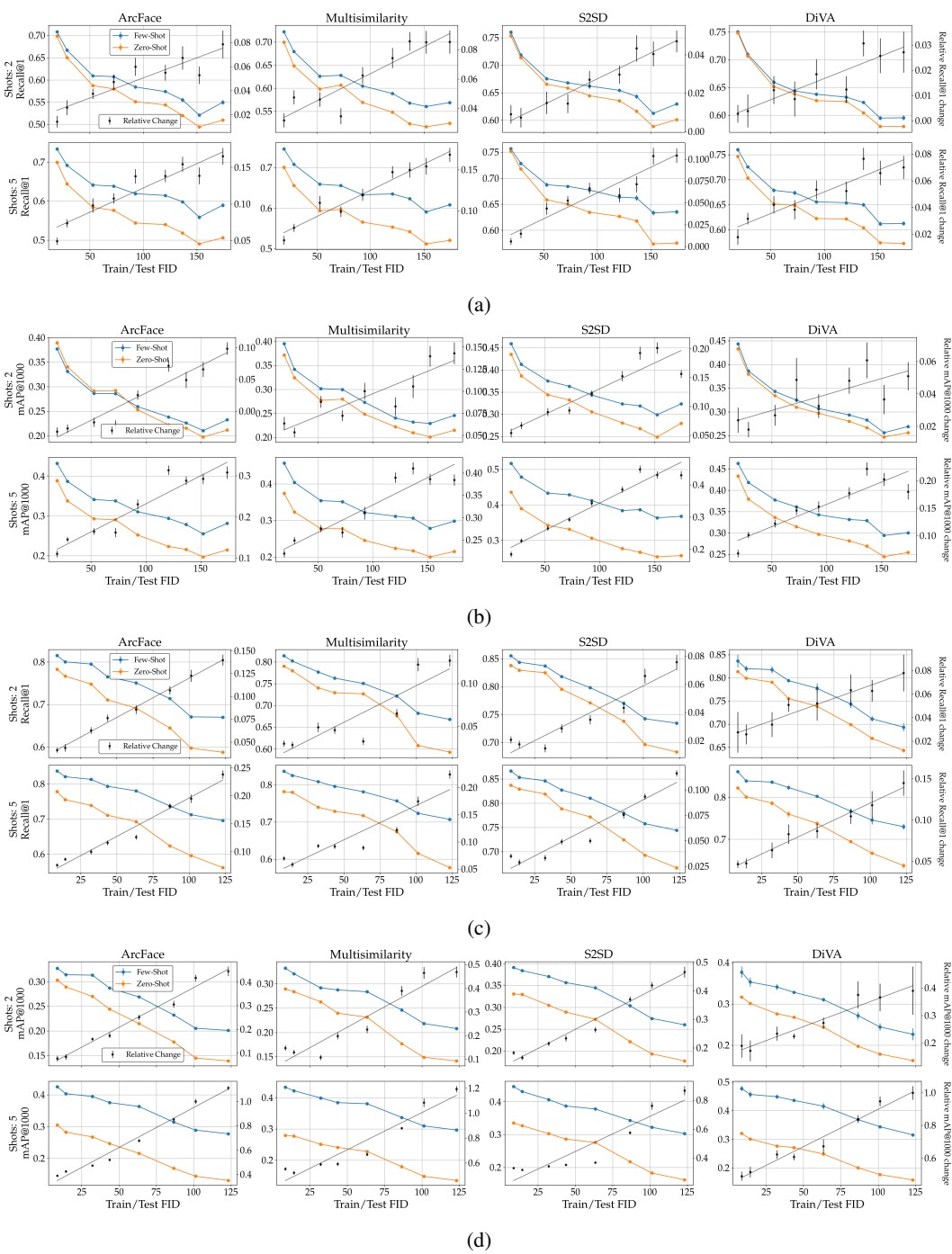

Figure 8: *Few-Shot adaptation of DML representations on CUB200-2011 and CARS196.* Columns show average Recall@1 performance over 10 episodes of 2- and 5-shot adaption as well as the baseline zero-shot DML results on the same train-test splits (based on *ooDML* benchmark) for various DML approaches (fewshot and zeroshot), highlighting a substantial benefit of few-shot adaptation for *a priori* unknown distribution shifts (see black line highlighting relative improvements). Relative improvements are computed as relative change of few-shot performance against respective zero-shot performance.

Table 7: Evaluation of zero-generalization and subsequent few-shot adaptation measured by Recall@1 and mAP@1000 based on few-shot dataplits built from the train-test splits of the *ooDML* benchmark (CUB200-2011). Results are further summarized in Fig. 8 (a) and (b).

| Metric | Shot | Use | Method↓ \| Split→ | 1 | 2 | 3 | 4 | 5 | 6 | 7 | 8 | 9 |
|---|---|---|---|---|---|---|---|---|---|---|---|---|
| R@1 | 2 | Zero | ArcFace | 69.88 ± 0.12 | 65.06 ± 0.12 | 58.76 ± 0.12 | 58.02 ± 0.10 | 55.14 ± 0.14 | 54.43 ± 0.11 | 52.04 ± 0.12 | 49.49 ± 0.13 | 51.00 ± 0.21 |
| | | | Multisimilarity | 70.02 ± 0.09 | 64.87 ± 0.11 | 59.88 ± 0.11 | 60.71 ± 0.17 | 56.96 ± 0.16 | 54.82 ± 0.18 | 52.33 ± 0.11 | 51.67 ± 0.14 | 52.45 ± 0.13 |
| | | | S2SD | 75.35 ± 0.15 | 71.38 ± 0.14 | 66.56 ± 0.14 | 65.85 ± 0.10 | 64.47 ± 0.14 | 63.56 ± 0.13 | 61.66 ± 0.12 | 58.87 ± 0.17 | 60.12 ± 0.10 |
| | | | DiVA | 74.86 ± 0.10 | 70.68 ± 0.16 | 65.17 ± 0.12 | 63.83 ± 0.19 | 62.63 ± 0.12 | 62.48 ± 0.10 | 60.45 ± 0.12 | 57.99 ± 0.16 | 57.96 ± 0.16 |
| | | Few | ArcFace | 70.88 ± 0.31 | 66.76 ± 0.39 | 60.97 ± 0.22 | 60.77 ± 0.35 | 58.45 ± 0.39 | 57.42 ± 0.34 | 55.55 ± 0.49 | 52.11 ± 0.33 | 55.01 ± 0.55 |
| | | | Multisimilarity | 72.26 ± 0.32 | 67.96 ± 0.29 | 62.64 ± 0.27 | 62.82 ± 0.29 | 60.52 ± 0.28 | 58.90 ± 0.32 | 56.82 ± 0.31 | 56.07 ± 0.37 | 56.92 ± 0.39 |
| | | | S2SD | 76.03 ± 0.33 | 71.90 ± 0.34 | 67.56 ± 0.36 | 66.81 ± 0.31 | 66.22 ± 0.27 | 65.45 ± 0.27 | 64.34 ± 0.40 | 61.26 ± 0.33 | 62.96 ± 0.33 |
| | | | DiVA | 75.07 ± 0.24 | 70.96 ± 0.43 | 65.96 ± 0.34 | 64.38 ± 0.40 | 63.79 ± 0.35 | 63.26 ± 0.32 | 62.30 ± 0.29 | 59.49 ± 0.36 | 59.54 ± 0.44 |
| | 5 | Zero | ArcFace | 69.96 ± 0.15 | 64.44 ± 0.12 | 58.43 ± 0.17 | 57.67 ± 0.21 | 54.42 ± 0.27 | 54.01 ± 0.28 | 51.85 ± 0.28 | 49.08 ± 0.26 | 50.61 ± 0.21 |
| | | | Multisimilarity | 70.15 ± 0.19 | 65.66 ± 0.19 | 59.42 ± 0.27 | 59.71 ± 0.13 | 56.59 ± 0.18 | 55.39 ± 0.19 | 54.26 ± 0.31 | 51.20 ± 0.21 | 52.10 ± 0.21 |
| | | | S2SD | 75.24 ± 0.22 | 71.81 ± 0.10 | 65.88 ± 0.22 | 65.03 ± 0.24 | 63.49 ± 0.27 | 62.69 ± 0.30 | 61.81 ± 0.22 | 57.40 ± 0.19 | 57.55 ± 0.18 |
| | | | DiVA | 74.71 ± 0.18 | 70.37 ± 0.17 | 65.18 ± 0.15 | 64.93 ± 0.11 | 62.27 ± 0.21 | 62.17 ± 0.22 | 60.40 ± 0.29 | 57.42 ± 0.26 | 57.26 ± 0.26 |
| | | Few | ArcFace | 73.41 ± 0.31 | 69.19 ± 0.32 | 64.16 ± 0.53 | 63.87 ± 0.34 | 61.92 ± 0.43 | 61.48 ± 0.33 | 59.86 ± 0.40 | 55.89 ± 0.49 | 58.98 ± 0.53 |
| | | | Multisimilarity | 74.66 ± 0.33 | 70.88 ± 0.26 | 65.95 ± 0.38 | 65.62 ± 0.35 | 63.36 ± 0.36 | 63.55 ± 0.35 | 62.40 ± 0.35 | 59.11 ± 0.45 | 60.88 ± 0.37 |
| | | | S2SD | 75.70 ± 0.21 | 72.88 ± 0.35 | 68.77 ± 0.41 | 68.46 ± 0.23 | 67.75 ± 0.29 | 66.38 ± 0.41 | 66.24 ± 0.54 | 63.37 ± 0.51 | 63.56 ± 0.43 |
| | | | DiVA | 76.05 ± 0.40 | 72.61 ± 0.25 | 67.93 ± 0.40 | 67.42 ± 0.46 | 65.59 ± 0.38 | 65.43 ± 0.38 | 65.02 ± 0.39 | 61.21 ± 0.45 | 61.28 ± 0.41 |
| mAP@1000 | 2 | Zero | ArcFace | 38.95 ± 0.08 | 34.03 ± 0.06 | 29.13 ± 0.04 | 29.22 ± 0.05 | 25.29 ± 0.04 | 22.25 ± 0.05 | 21.56 ± 0.05 | 19.76 ± 0.04 | 21.17 ± 0.05 |
| | | | Multisimilarity | 37.19 ± 0.07 | 32.46 ± 0.07 | 27.75 ± 0.04 | 27.99 ± 0.04 | 24.86 ± 0.03 | 22.22 ± 0.10 | 21.00 ± 0.06 | 20.12 ± 0.06 | 21.51 ± 0.05 |
| | | | S2SD | 43.56 ± 0.09 | 38.66 ± 0.07 | 34.41 ± 0.05 | 33.23 ± 0.05 | 30.59 ± 0.04 | 28.08 ± 0.05 | 26.76 ± 0.06 | 24.89 ± 0.04 | 27.98 ± 0.04 |
| | | | DiVA | 43.33 ± 0.08 | 37.99 ± 0.05 | 33.43 ± 0.04 | 31.00 ± 0.05 | 29.75 ± 0.03 | 28.02 ± 0.07 | 26.69 ± 0.06 | 24.73 ± 0.05 | 25.63 ± 0.05 |
| | | Few | ArcFace | 37.71 ± 0.25 | 33.11 ± 0.21 | 28.63 ± 0.21 | 28.60 ± 0.19 | 25.94 ± 0.18 | 23.84 ± 0.20 | 22.62 ± 0.27 | 21.06 ± 0.23 | 23.26 ± 0.20 |
| | | | Multisimilarity | 39.57 ± 0.27 | 34.20 ± 0.16 | 30.20 ± 0.18 | 30.02 ± 0.17 | 27.36 ± 0.23 | 24.07 ± 0.20 | 23.22 ± 0.27 | 22.94 ± 0.23 | 24.59 ± 0.26 |
| | | | S2SD | 45.94 ± 0.31 | 41.26 ± 0.22 | 37.53 ± 0.23 | 36.33 ± 0.20 | 34.35 ± 0.14 | 32.37 ± 0.24 | 31.91 ± 0.29 | 29.90 ± 0.24 | 32.36 ± 0.19 |
| | | | DiVA | 44.36 ± 0.33 | 38.67 ± 0.17 | 34.33 ± 0.21 | 32.51 ± 0.42 | 30.72 ± 0.22 | 29.37 ± 0.21 | 28.31 ± 0.29 | 25.63 ± 0.22 | 26.94 ± 0.22 |
| | 5 | Zero | ArcFace | 38.91 ± 0.10 | 33.79 ± 0.05 | 29.30 ± 0.06 | 29.06 ± 0.08 | 25.24 ± 0.07 | 22.38 ± 0.09 | 21.62 ± 0.09 | 19.75 ± 0.05 | 21.52 ± 0.06 |
| | | | Multisimilarity | 37.45 ± 0.09 | 32.38 ± 0.06 | 27.80 ± 0.06 | 27.76 ± 0.09 | 24.61 ± 0.07 | 22.46 ± 0.08 | 21.75 ± 0.10 | 20.10 ± 0.06 | 21.56 ± 0.07 |
| | | | S2SD | 43.61 ± 0.12 | 39.01 ± 0.08 | 34.25 ± 0.07 | 33.16 ± 0.09 | 30.67 ± 0.07 | 27.73 ± 0.09 | 26.70 ± 0.12 | 25.40 ± 0.07 | 25.74 ± 0.12 |
| | | | DiVA | 43.34 ± 0.10 | 37.96 ± 0.08 | 33.65 ± 0.07 | 31.49 ± 0.12 | 29.72 ± 0.08 | 28.17 ± 0.10 | 26.95 ± 0.12 | 24.47 ± 0.09 | 25.47 ± 0.09 |
| | | Few | ArcFace | 43.22 ± 0.28 | 38.71 ± 0.20 | 34.14 ± 0.22 | 33.79 ± 0.32 | 31.06 ± 0.29 | 29.41 ± 0.32 | 27.86 ± 0.20 | 25.53 ± 0.25 | 28.15 ± 0.32 |
| | | | Multisimilarity | 45.68 ± 0.28 | 40.45 ± 0.24 | 35.46 ± 0.21 | 35.18 ± 0.31 | 32.25 ± 0.34 | 31.21 ± 0.24 | 30.69 ± 0.26 | 27.88 ± 0.24 | 29.86 ± 0.26 |
| | | | S2SD | 51.67 ± 0.35 | 47.86 ± 0.25 | 43.33 ± 0.22 | 42.85 ± 0.19 | 41.21 ± 0.24 | 38.43 ± 0.25 | 38.68 ± 0.26 | 36.35 ± 0.26 | 36.82 ± 0.28 |
| | | | DiVA | 46.29 ± 0.27 | 41.82 ± 0.20 | 37.77 ± 0.22 | 36.10 ± 0.24 | 34.27 ± 0.28 | 33.17 ± 0.28 | 32.93 ± 0.29 | 29.44 ± 0.27 | 30.07 ± 0.35 |

Table 8: Evaluation of zero-generalization and subsequent few-shot adaptation measured by Recall@1 and mAP@1000 based on few-shot dataplits built from the train-test splits of the *ooDML* benchmark (CARS196). Results are further summarized in Fig. 8 (c) and (d).

| Metric | Shot | Use | Method↓ \| Split→ | 1 | 2 | 3 | 4 | 5 | 6 | 7 | 8 |
|---|---|---|---|---|---|---|---|---|---|---|---|
| R@1 | 2 | Zero | ArcFace | 78.30 ± 0.08 | 76.67 ± 0.09 | 74.82 ± 0.08 | 71.11 ± 0.08 | 69.18 ± 0.08 | 64.53 ± 0.12 | 59.79 ± 0.09 | 58.82 ± 0.09 |
| | | | Multisimilarity | 79.07 ± 0.07 | 78.02 ± 0.09 | 74.10 ± 0.06 | 72.96 ± 0.11 | 72.72 ± 0.06 | 67.80 ± 0.17 | 60.82 ± 0.14 | 59.31 ± 0.13 |
| | | | S2SD | 83.84 ± 0.06 | 82.96 ± 0.09 | 82.53 ± 0.08 | 79.56 ± 0.07 | 77.14 ± 0.10 | 73.84 ± 0.09 | 69.67 ± 0.13 | 68.31 ± 0.11 |
| | | | DiVA | 81.39 ± 0.20 | 79.94 ± 0.08 | 79.08 ± 0.35 | 75.58 ± 0.28 | 73.89 ± 0.18 | 69.94 ± 0.31 | 66.97 ± 0.23 | 64.36 ± 0.48 |
| | | Few | ArcFace | 81.54 ± 0.21 | 80.03 ± 0.30 | 79.53 ± 0.26 | 76.54 ± 0.26 | 75.10 ± 0.31 | 71.42 ± 0.26 | 67.13 ± 0.38 | 67.05 ± 0.34 |
| | | | Multisimilarity | 81.45 ± 0.28 | 80.25 ± 0.27 | 77.74 ± 0.41 | 76.31 ± 0.29 | 75.10 ± 0.28 | 72.23 ± 0.29 | 68.25 ± 0.44 | 66.84 ± 0.38 |
| | | | S2SD | 85.57 ± 0.20 | 84.41 ± 0.20 | 83.74 ± 0.21 | 81.85 ± 0.23 | 79.85 ± 0.23 | 77.04 ± 0.28 | 74.27 ± 0.34 | 73.48 ± 0.35 |
| | | | DiVA | 83.65 ± 1.39 | 82.02 ± 0.66 | 81.79 ± 0.75 | 79.43 ± 0.33 | 77.76 ± 1.06 | 74.38 ± 0.88 | 71.19 ± 0.57 | 69.38 ± 0.86 |
| | 5 | Zero | ArcFace | 77.82 ± 0.14 | 75.52 ± 0.11 | 73.91 ± 0.12 | 71.10 ± 0.16 | 69.30 ± 0.16 | 62.40 ± 0.12 | 59.64 ± 0.15 | 56.20 ± 0.18 |
| | | | Multisimilarity | 78.19 ± 0.15 | 78.00 ± 0.14 | 74.00 ± 0.18 | 72.93 ± 0.16 | 71.77 ± 0.17 | 67.44 ± 0.19 | 61.59 ± 0.15 | 57.77 ± 0.18 |
| | | | S2SD | 83.75 ± 0.11 | 82.96 ± 0.13 | 81.92 ± 0.13 | 78.88 ± 0.12 | 77.19 ± 0.09 | 72.47 ± 0.19 | 69.30 ± 0.10 | 66.66 ± 0.09 |
| | | | DiVA | 82.27 ± 0.10 | 80.12 ± 0.31 | 78.65 ± 0.56 | 76.03 ± 0.67 | 73.82 ± 0.53 | 69.46 ± 0.26 | 66.72 ± 0.43 | 63.76 ± 0.63 |
| | | Few | ArcFace | 83.74 ± 0.19 | 82.07 ± 0.19 | 81.30 ± 0.33 | 79.34 ± 0.25 | 78.04 ± 0.25 | 73.75 ± 0.26 | 71.27 ± 0.40 | 69.60 ± 0.36 |
| | | | Multisimilarity | 83.67 ± 0.28 | 82.56 ± 0.35 | 80.87 ± 0.20 | 79.63 ± 0.29 | 78.17 ± 0.26 | 75.68 ± 0.32 | 72.37 ± 0.52 | 70.73 ± 0.39 |
| | | | S2SD | 86.69 ± 0.17 | 85.37 ± 0.21 | 84.65 ± 0.19 | 82.77 ± 0.18 | 81.05 ± 0.18 | 77.96 ± 0.18 | 75.79 ± 0.17 | 74.43 ± 0.18 |
| | | | DiVA | 86.11 ± 0.36 | 83.94 ± 0.35 | 83.64 ± 0.45 | 82.35 ± 0.52 | 80.24 ± 0.31 | 76.73 ± 0.56 | 74.61 ± 1.02 | 73.02 ± 0.67 |
| mAP@1000 | 2 | Zero | ArcFace | 30.37 ± 0.04 | 28.98 ± 0.03 | 27.03 ± 0.02 | 24.46 ± 0.03 | 21.52 ± 0.03 | 17.81 ± 0.05 | 14.52 ± 0.03 | 13.93 ± 0.03 |
| | | | Multisimilarity | 28.95 ± 0.03 | 28.38 ± 0.02 | 26.33 ± 0.02 | 23.98 ± 0.04 | 23.13 ± 0.03 | 17.71 ± 0.05 | 14.90 ± 0.03 | 14.17 ± 0.04 |
| | | | S2SD | 33.08 ± 0.04 | 32.96 ± 0.04 | 30.49 ± 0.02 | 28.92 ± 0.05 | 27.28 ± 0.03 | 22.16 ± 0.05 | 19.33 ± 0.04 | 17.74 ± 0.05 |
| | | | DiVA | 31.69 ± 0.11 | 30.13 ± 0.08 | 27.64 ± 0.03 | 26.80 ± 0.10 | 24.38 ± 0.12 | 19.77 ± 0.22 | 17.89 ± 0.12 | 16.32 ± 0.13 |
| | | Few | ArcFace | 32.75 ± 0.35 | 31.43 ± 0.35 | 31.35 ± 0.16 | 28.74 ± 0.27 | 26.93 ± 0.24 | 23.27 ± 0.25 | 20.58 ± 0.19 | 20.12 ± 0.28 |
| | | | Multisimilarity | 33.23 ± 0.32 | 32.08 ± 0.29 | 29.19 ± 0.29 | 28.75 ± 0.32 | 28.38 ± 0.36 | 24.63 ± 0.36 | 21.85 ± 0.36 | 20.82 ± 0.32 |
| | | | S2SD | 39.14 ± 0.22 | 38.44 ± 0.27 | 37.06 ± 0.25 | 35.69 ± 0.31 | 34.50 ± 0.25 | 30.38 ± 0.20 | 27.46 ± 0.22 | 26.01 ± 0.33 |
| | | | DiVA | 37.71 ± 1.35 | 35.32 ± 1.14 | 34.11 ± 0.71 | 32.83 ± 0.23 | 31.05 ± 0.46 | 27.19 ± 0.93 | 24.44 ± 0.88 | 22.68 ± 1.42 |
| | 5 | Zero | ArcFace | 30.55 ± 0.08 | 28.28 ± 0.06 | 26.78 ± 0.05 | 24.68 ± 0.06 | 21.64 ± 0.04 | 16.94 ± 0.05 | 14.46 ± 0.04 | 13.14 ± 0.06 |
| | | | Multisimilarity | 27.95 ± 0.07 | 27.78 ± 0.07 | 25.15 ± 0.06 | 24.16 ± 0.07 | 22.83 ± 0.04 | 17.95 ± 0.04 | 14.88 ± 0.05 | 13.55 ± 0.05 |
| | | | S2SD | 33.57 ± 0.06 | 32.71 ± 0.09 | 30.29 ± 0.07 | 28.64 ± 0.06 | 27.61 ± 0.05 | 21.78 ± 0.05 | 18.30 ± 0.19 | 16.22 ± 0.22 |
| | | | DiVA | 32.06 ± 0.13 | 30.14 ± 0.10 | 27.67 ± 0.12 | 27.13 ± 0.22 | 24.87 ± 0.07 | 20.05 ± 0.11 | 17.66 ± 0.19 | 15.79 ± 0.22 |
| | | Few | ArcFace | 42.52 ± 0.21 | 40.33 ± 0.31 | 39.46 ± 0.17 | 37.55 ± 0.21 | 36.33 ± 0.28 | 31.42 ± 0.20 | 28.92 ± 0.28 | 27.76 ± 0.16 |
| | | | Multisimilarity | 43.43 ± 0.37 | 42.30 ± 0.38 | 39.99 ± 0.33 | 38.49 ± 0.35 | 38.12 ± 0.31 | 33.75 ± 0.19 | 31.02 ± 0.51 | 29.71 ± 0.31 |
| | | | S2SD | 44.67 ± 0.12 | 43.13 ± 0.16 | 40.66 ± 0.15 | 38.72 ± 0.17 | 37.80 ± 0.14 | 34.31 ± 0.24 | 32.24 ± 0.33 | 30.30 ± 0.30 |
| | | | DiVA | 47.79 ± 0.85 | 45.72 ± 1.00 | 44.92 ± 0.72 | 43.62 ± 0.42 | 41.62 ± 1.09 | 36.89 ± 0.43 | 34.41 ± 0.37 | 31.56 ± 0.49 |