# OpenReview forum: "Characterizing Generalization under Out-Of-Distribution Shifts in Deep Metric Learning"
_NeurIPS.cc/2021/Conference — NeurIPS 2021 Poster_

### Official Review · Reviewer_de3V · 2021-07-14

**Rating:** 7
**Confidence:** 4

**Summary:**

This work proposes a new way to measure OOD performance of metric-learning algorithms. It introduces a proxy-metric for measuring ranking dataset generalization difficulty based on FID score. They construct several train/test splits of increasing difficulty based on this score. They show that splits constructed in such a way are well correlated with performance, thus validating the proposed proxy-metric of OOD generalization difficulty.  Several experiments are performed to assess the performance of modern metric learning methods. An AUC type score over all splits (difficulties) is proposed as an overall score for various methods. Finally, additional experiments that include increasing network capacity, using self-supervised pre-trained models, and fine-tuning with few-shot learning are presented.

**Limitations And Societal Impact:**

Because this work presents results on three particular datasets, it is not clear if the methods will generalize to others, though the results of the paper suggest that they will.

**Main Review:**

The proposed proxy-metric of difficulty makes sense, and is nicely empirically validated. I am glad to see that the splits/code will be open sourced because these will be very valuable to the community. Particularly moving beyond the recent negative results  [52, 40].
Because this paper presents a new benchmark it may be well suited to the Benchmarks & Datasets track at NeurIPS (although I am happy to have it accepted via the main submission route as well). Although some of the conclusions are not particularly surprising (e.g. few-shot learning on the target distribution helps, self-supervised learning doesn't always work), they are nevertheless useful experiments to the community.
Some questions:
1-How were the error bars in Figure 3 computed, aren't the splits deterministic?
2-According to Figure 1 there is a wide, but limited region for each dataset where swapping affects the FID. Do you only use these dataset splits in your experiments as the early/late ones lead to saturated FID scores ? I know that these are further extended using removals.
3- Would the methods presented in the paper have impact on the few-shot learning community as well (e.g. MetaDataset uses fixed splits)?
4-It would be nice to see other methods that are known to increase generalization i the analyses, data augmentation is an obvious one that comes to mind.

**Time Spent Reviewing:**

2

---

> ### Author Response · Authors · 2021-08-09
> **Response to Reviewer de3V**
>
> We thank the reviewers for their thorough reviews and helpful comments, as well as for appreciating the writing, helpfulness of figures, ease of understanding and overall quality of the paper. In addition, we thank the reviewers for understanding the usefulness, impact, novelty and relevance of our research.
>
> #### **Re. Expected benefit of Few-shot DML:**
> Please see our response "Few-shot Learning helps DML" to reviewer VKkF.
>
> #### **Re. Errorbars Figure 3:**
> While the train-test splits in our ooDML benchmark are deterministic and fixed, the initialization of the network weights of the evaluated models is not. The error bars indicate the standard deviation of performance over multiple runs using different random seeds.
>
> #### **Re. Saturation of FID/Usage of other splits:**
> As noted correctly, FID scores saturate when relying solely on swapping - this saturation is reflected both in our iterative process getting stuck in swapping the same set of samples back and forth, but also in very similar performance (as FIDs are very similar), and thus, no additional insights would be gained using splits from the saturated area. This is one of the reasons why we subsequently also resort to removal of classes to artificially increase the difficulty of these splits, however, only to the point where at least 50% of the initial data is still retained (L150).
>
> #### **Re. Application for Few-Shot Learning:**
> Specifically for few-shot learning, our procedure for generating data splits could certainly be applied to introduce more semantically difficult train-validation-test splits covering learning problems of varying difficulty, and hence would tie in well with related research highlighted by VKkF.
>
> #### **Re. Other methods:**
> We agree with the reviewer that evaluating more general methods for increasing representation generalization could be very interesting. However, the primary focus of this work is to identify and analyze the issue of single, fixed train-test splits for evaluating OOD generalization and offering a solution with our ooDML benchmark. Moreover, we focused in particular on whether conceptual differences in Deep Metric Learning approaches impact OOD generalization, as methods such as data augmentation apply similarly to all DML approaches. In addition, recent research such as Taori et al. 2020 ("Measuring Robustness to Natural Distribution Shifts in Image Classification", arxiv.org/abs/2007.00644) has shown that data augmentation strategies may be insufficient to handle multiple natural distribution shifts, further supporting more detailed research into general DML models under realistic shift settings instead.

---

> > ### Comment · Reviewer_de3V · 2021-08-17
> > **Reviewer response to author response**
> >
> > I acknowledge the authors' response and maintain my decision to accept.

---

### Official Review · Reviewer_VKkF · 2021-07-16

**Rating:** 7
**Confidence:** 4

**Summary:**

The authors present a new benchmark for deep metric learning algorithms, particularly for zero-shot tasks, in order to reflect their generalization under out-of-distribution shifts performance. They propose a novel way of splitting train and test data with increasing difficulty(the gap(distribution shift) between train and test get higher). FID score is chosen to determine the distribution shift between train and test. After an initial split is chosen, the classes are exchanged between the train and test data to obtain a higher FID measure at each time iteratively. The authors suggest to use ROC curve to obtain a one single criteria, while keeping the results for each split in order to observe the performance based on task difficulty. They include experimental analysis on their proposed criteria and show results for some existing methods and architectures. The authors also analyze query-support framework in few-shot learning and its effect on the generalization performance.

**Limitations And Societal Impact:**

Proposing a benchmark is always important in fairness in the evaluation of the algorithms. The authors already mention weak points.

**Main Review:**

#### *Originality:*

 -  There are a few recent previous attempts to analyze quantify the difficulty of few/zero shot tasks and the it is still an important open question. Progressively increasing the difficulty of train-test split and obtaining a range of tasks as a benchmark is novel as of my knowledge. They used Resnet FID distance to achieve this in order to cover many of the existing methods.
    - The related work contains sufficient citations of the previous contributions(especially 52,40,27), however the authors could explain them in a couple of more sentences to create a better set up for the current progress and their contribution.
    - Suggested citations: Two Sides of Meta-Learning Evaluation:In vs. Out of Distribution(Amrith Setlur et. al.). Huang, Gabriel, Hugo Larochelle, and Simon Lacoste-Julien. "Are Few-shot Learning Benchmarks Too Simple?." (2019).

 - The paper also switches from zero shot case to few shot by including some class samples into their test split and show that the generalization improves (as expected) as another novel contribution.

#### *Quality*

- Choosing FID metric can be unreliable as the authors are aware of. Figure S2 is sufficient proof of concept for me at this stage. Although a comparison between the datasets might not be reliable(Table 1 suggests the otherwise), it can still be a relative measure for a given fixed dataset.


- Why didn't you simultaneously decrease FID to obtain simpler/simplest tasks given that your initialization is random? Is it guaranteed that random initialization is simple enough to cover a wider difficulty range?


- The methods seem to act similar in recall criteria(Figure 2), is it necessary to use splitting via FID in this case? The authors could provide better insights on why mAP and recall behaves very differently(Figure 2).


- The authors mention their weaknesses in having similar max FID value for different datasets, being flawed in its descriptiveness(Figure 4) and provided sufficient explanations for the possible reasons.


- Can this method be used in domain adaptation? The method might be costly for the large datasets, isn't?


- Adapting the method directly might cause a bias for example if minimizing FID measure alone helps to get better results in more difficult data splits.

#### *Clarity:*

- The paper is clearly written, the figures are very helpful.


 - As I mentioned above, the related work section could be more informative.

#### *Significance:*

- The proposed method is applicable to any zero/few shot dataset(as long as not very large and checking FID scores before using as a benchmark might be needed). This makes the method widely useable.

- The problem itself is important and hard to solve, I like the idea of creating different distribution shifts for a given dataset. The method has a potential to be further improved or inspire future works, both experimental or theoretical approaches.





**Time Spent Reviewing:**

3

---

> ### Author Response · Authors · 2021-08-09
> **Response to Reviewer VKkF**
>
> We thank the reviewers for their thorough reviews and helpful comments, as well as for appreciating the writing, helpfulness of figures, ease of understanding and overall quality of the paper. In addition, we thank the reviewers for understanding the usefulness, impact, novelty and relevance of our research.
>
> #### **Re. Missing citations \& expanding Related Works:**
> We thank the reviewer for highlighting additional contributions in this research direction and have included them in our Related Works section. In comparison, our work is the first to tackle entire spectra of OOD shifts in the domain of zero-shot/Deep Metric Learning and proposes a novel protocol to generate new benchmarks with realistic appeal. Moreover, we also study the impact of different conceptual approaches on OOD generalization, the consistency of generalization metrics, as well as the introduction of the not yet explored field of few-shot DML as a reliable mean to improve OOD generalization, especially for increasingly more difficult train-test splits.
> Given that metric-based methods are common in few-shot learning, future research can be done linking both fields of research.
>
> #### **Re. Few-Shot Learning helps DML.**
> We thank the reviewer for appreciating the purpose of these experiments. Indeed, the performance gain due to having access to limited samples from the test distribution is not surprising. However, it is nevertheless important to quantify the actual benefit and no work (to the best of our knowledge) has investigated the impact of few-shot DML. To this end, we make the interesting observation that few-shot DML becomes disproportionally more beneficial as train-test shifts get larger.
>
> #### **Re. FID as metric:**
> As the reviewer has correctly pointed out, measuring true OOD shift between two data distributions is difficult and FID may not be the best metric to measure distribution shifts. However, as noted in our reply to vS38 and as the reviewer correctly pointed out, we have included notable experimental support for this metric both qualitatively (UMAP of increasingly more OOD splits, Fig. S2) and quantitatively (consistency across models and pretrainings, Fig. S1). Finally, the usage of FID in our work only serves the purpose of being able to infer relative differences between data splits drawn from a given dataset, which is strongly supported by our experiments (e.g. Fig. 1, Fig. 2, Fig. S1, Fig. S2). As such, potential inconsistencies in absolute values do not interfere with the purpose and goals of this work.
>
> #### **Re. Decreasing FID:**
> We actually did exactly that for the CUB200-2011 benchmark (see Fig. 1 and 2). For the other two benchmarks sets (CARS196, SOP), we found that data splits could not be made significantly "easier" as can be seen in Fig. 1, where the black line denotes the difficulty of the standard split. Hence, we resort in these cases to the default splits as our "easiest" ones.
>
> #### **Re. Different results on mAP:**
> The bottom row plots in Fig. 2 actually show relative changes between the Recall@1 values in the top row. For mAP, the aggregated equivalent to Fig. 3 can be found in the supplementary, as well as the exact values for the shift progressions in Tab. S4 - S6 corresponding to progressions as found in Fig. 2. For completeness, we have explicitly added the mAP-equivalent to Fig. 2 in the supplementary which, as given in Tab. S4 - S6, behaves very similar to the Recall@1 scores in Fig. 2.
>
> #### **Re. Necessity of data splitting in case of Fig.2:**
> Evaluation and comparison based on increasingly more difficult train-test splits is important to measure generalization in a more realistic setting. In Fig. 2, we observe for example that the ranking between methods can actually change quite dramatically (see e.g. on SOP, L198-206), where previous SOTA methods actually perform much worse in more challenging settings. Moreover, we see that difference between methods become more prominent, which goes against the lately observed performance saturation in DML on the commonly used default train-test splits (e.g. Roth et al. 2020, Musgrave et al. 2020). To construct a series of increasingly more difficult train-test splits, i.e. ooDML, we have to be able to measure the extent of the distribution gap.
>
> #### **Re. Comparison between datasets:**
> Comparison of absolute values between datasets may not be meaningful as also pointed out by the reviewer. Hence, FID values should indeed be treated more as a distribution shift metric within a given domain/dataset. We have highlighted this better in Sections 3.2 and 3.3.
>
> #### **Re. Adapting the method directly might cause a bias for example if minimizing FID measure alone helps to get better results in more difficult data splits:**
> Our method for creating data splits of increasing difficulty yielding the *ooDML* benchmark is independent of training the actual models. Our data splits are chosen to approximately uniformly cover the spectrum of OOD shift and are fixed, hence being identical for all models to be evaluated. As for each split, no access to the test data is given during training, there is no way for a method to directly minimize for the train/test FID score.
>
> #### **Re. Can this method be used in domain adaptation?:**
> Our approach can potentially be utilized in other areas of out-of-distribution generalization research such as few-shot learning. In case of domain adaptation, the class-switching procedure of our protocol would require to swap classes between source and target domain which typically differ in a general property (e.g. MNIST dataset to SVHN dataset, i.e. black and white numbers vs. colored house numbers). Hence, swapping samples would interfere with the actual motivation behind domain adaption, as colored house numbers from the a priori unknown target domain would now also be found in the training dataset. In contrast, standard zero-shot and few-shot classification tasks operate withing the same domain, and only assume differences on a class- instead of a domain-level.

---

### Official Review · Reviewer_vS38 · 2021-07-16

**Rating:** 7
**Confidence:** 3

**Summary:**

The authors of this paper investigate how generalization is affected by varying the training and testing data splits in deep metric learning (DML). Specifically, they create train/test splits (splitting on classes) which increase in difficulty (meaning that the distributions are more different) and evaluate DML generalization. To measure the gap between train and test distributions, they use FID with ResNet-50. Experimentally, they use popular benchmarks and the Recall@k metric to show that performance decreases as the train/test FID distances become larger. This, and other experiments, show that using a fixed train/test split can lead to misleading conclusions about generalization.

**Ethical Concerns:**

Nope

**Limitations And Societal Impact:**

Limitations:

The main limitation I see is the application of a mean-only FID using ResNet-50 instead of Inception-v3 on different datasets than ImageNet.

Potential negative societal impacts:

None that I can think of.

**Main Review:**

Originality:

The idea is very interesting and original. It makes sense that different train/test splits will make the training harder and this should be investigated.

Quality:

Overall, the paper is of high quality and is relatively easy to understand.

There are some mistakes in the paper. For example, on line 107, the authors state that FID uses Inception-v2 when it actually is Inception-v3.

Some things are not well justified. For example, the approximation of FID using only the mean is not well justified. Just because the mean-FID is monotonically increasing per iteration does not mean that the FID will behave the same. More justification is needed here. Another justification that is needed is why ResNet-50 was used and why the authors think that FID with this network will work well on the CUB200-2011, CARS196, and Stanford Online Products datasets, assuming that it was trained on ImageNet.

There might be an issue with the FID calculation as well. If the train/test splits have varying (or small) sizes then FID will become biased. See “Effectively Unbiased FID and Inception Score and where to find them” for more details.

Clarity:

Overall, the paper is easy to read and understand.

The fonts in the figures are too small, making them hard to read. The captions and descriptions are also hard to understand. Moreover, several figures do not print well in black and white.

Significance:

I am not sure about the significance, because I don’t specialize in deep metric learning. However, the authors make a great point that if the train/test splits make a huge difference, then they should be investigated. However, the authors also split on classes, which may be causing the huge difference in performance.

**Time Spent Reviewing:**

3

---

> ### Author Response · Authors · 2021-08-09
> **Response to Reviewer vS38**
>
> We thank the reviewers for their thorough reviews and helpful comments, as well as for appreciating the writing, helpfulness of figures, ease of understanding and overall quality of the paper. In addition, we thank the reviewers for understanding the usefulness, impact, novelty and relevance of our research.
>
> #### **Re. Mistakes and figure readability:**
> We thank the reviewer for pointing out these issues and have fixed them for the camera-ready version. Specifically, this includes:
> * Iv3 instead of Iv2.
> * Fontsize of figures increased.
> * Improved contrast in figures for better readability.
>
> #### **Re. Justification of our FID approximation as a measure to increase semantic OOD shifts:**
> [1] *Approximation of FID by mean.*
> We use the class-mean FID approximation only for identifying classes to be swapped or removed -  as we care about OOD shifts caused by semantic changes rather than changes driven by more general, e.g. background/class-unrelated information/biases, we ground our splitting routine on assignment of whole classes to train and test sets.
> This best maintains the standard evaluation setting in DML (e.g. Wu et al. 2017, Roth et al. 2020, Musgrave et al. 2020) aiming at correctly relating samples from novel, unseen classes.
> Given these reasons, a class-mean based approximation to FID is both sufficient and well aligned with the goal of our iterative split generation.
> Moreover, this approach is further supported by previous works (e.g. Lin et al. 2018, Roth et al. 2019) showcasing unimodality of in-class sample distribution to be a reasonable approximation. In addition, Fig. 1 and Fig. S2 show that through iterative optimization of our class-mean approximation to FID, class-mean FID scores are indeed increasing (Fig. 1) while also gradually increasing the OOD shifts on a per-sample basis (Fig. S2). Finally, this approximation of FID is robust and consistent across pretraining methods and backbone architectures on a relative level (Fig. S1), which indicates the validity of this metric to produce quantifiable harder splits. We thank the reviewer for highlighting the missing justification in the main text and have extended Section 3.1 to include this information.
>
> [2] *Imagenet pretrained ResNet-50 instead of Iv3.*
>  Our work utilizes an ImageNet pretrained ResNet50 as this is the de-facto standard architecture in DML (L112), allowing for consistency with prior research. For the paper, to allow the reader to better understand the conceptual idea, we overloaded the term FID (L113) to refer to Wasserstein distances using a ResNet-50. As we do not care about absolute, but relative scores for the extent of OOD shifts within data splits (thus not being affected by biases as noted in Chong et al. 2019),  Fig. S1 provides additional support for our choice of metric, as across models and pretraining methods/datasets, the relative ranking based on our classmean FID metric is retained, thus allowing us to stick to a ResNet50 backbone for consistency. Finally, Fig. S2 showcases qualitatively that our definition of FID is sufficient in creating increasingly more difficult data splits.
>
> [3] *Impact of train/test split size of FID bias.*
> We have extended our Related Works to address the mentioned reference: While Chong et al. 2019 highlight that FID on smaller sample sizes are not correct reflections of true FID values, in our case, absolute values do not matter as much. Instead, we care about relative differences between splits to quantify the difficulty of our generated splits with respect to the default ones. In addition, supplementary section C and Table S2 provide limit cases for semantic splits, giving empirical upper bounds to offer some alignment of scores.
> Finally, we find that the relative ordering, which is most crucial to us, is fully retained across models and pretraining setups (Fig. S1).
>
> #### **Re. Significance of work:**
> Our work is significant as we identified and analyzed crucial deficiencies and shortcomings of the prevailing zero-shot study protocols in DML while also providing a novel testbed for research to investigate these issues going forward. Moreover, as also highlighted by reviewer de3V, our work shows that the reported issue of performance saturation in DML (e.g. Roth et al. 2020, Musgrave et al. 2020) is less severe when probing learning problems that are more difficult than the standard benchmark data splits. This indeed concludes that DML methods should be evaluated across a spectrum of train/test shifts and future research in DML will benefit from benchmarks such as ooDML. In addition, our insights can also transfer to other areas targeting the important topic of OOD generalization (as discussed in our response to kgM7), such as few-shot learning (see reviewer VKkF) and provides a stepping stone for subsequent research to build upon.
> Finally, we offer quantitative support for potential future research into so far unstudied few-shot DML as a reliable remedy for OOD generalization improvements especially when test shifts are not known a priori.

---

> > ### Comment · Reviewer_vS38 · 2021-08-24
> > **Thank you for the response**
> >
> > Thank you for the response! You helped clear up some confusion that I had. I will maintain my original score as it is a good paper.

---

### Official Review · Reviewer_kgM7 · 2021-07-26

**Rating:** 6
**Confidence:** 3

**Summary:**

This paper proposed a new benchmark for deep metric learning under varying degree of distribution shift. First, it introduces a method for creating train/test split with increasing FID --- thereby increasing distribution shift between train and test. Second, various metric learning methods are compared under the benchmarks of varying difficulties.

**Limitations And Societal Impact:**

Limitations or potential negative societal impact are not included in this paper. Including limitations as well as the position of this paper against prior literature will greatly enhance the quality of this paper.

**Main Review:**

[Strengths]
- It is interesting and novel to quantify (a) the degree of distribution shift & (b) the effect of such shift.
- The paper is largely well-written and easy to understand.

[Weaknesses]
- It introduces the problem, but it does not tell much about the problem. From machine learning theory, we know that out-of-distribution benchmark is hard. Introducing a simple method to mitigate the problem can be a plus to this paper.
- It does not provide much insight on why different method performs better or worse under out-of-distribution scenarios.
- I'm not sure how useful this benchmark is. We know that machine learning does not aim to work under out-of-distribution shift. Also, I think this benchmark is meaningful when the ranking between methods under i.i.d benchmark is different from out-of-distribution benchmark. From what I observed, the relative ranking is largely preserved under different difficulties of data split.

**Time Spent Reviewing:**

1.5

---

> ### Author Response · Authors · 2021-08-09
> **Response to Reviewer kgM7**
>
> We thank the reviewers for their thorough reviews and helpful comments, as well as for appreciating the writing, helpfulness of figures, ease of understanding and overall quality of the paper. In addition, we thank the reviewers for understanding the usefulness, impact, novelty and relevance of our research.
>
> #### **Re. Not telling enough about the problem:**
> As mentioned in e.g. L19-25 \& L31-34, we highlight why we care about the problem: similarity-based (e.g. retrieval) applications in Deep Metric Learning are commonly evaluated on classes not encountered before during training in a zero-shot setting. This requires learning metric spaces which transfer well to classes not known a priori, requiring a certain robustness to semantic shifts. Unfortunately, existing evaluation protocols insufficiently capture how DML methods perform in such settings (L90-103), with current benchmarks providing only a single degree of difficulty, which in addition is semantically closer to i.i.d. train-test splits (L114-125). This setting is disconnected from realistic application settings, where one needs to know how different methods behave under different train-test shifts (L90-96). This issue is exacerbated with recent research (e.g. Roth et al. 2020, Musgrave et al. 2020) showcasing a high degree of performance saturation on these default splits, pointing to the need for new benchmarks that provide both a more complex and meaningful study setting for researchers to investigate benefits of different conceptual approaches to DML (e.g. L40-44).
>
> #### **Re. Insight into why different methods perform better or worse:**
> While we do provide conceptual insights as to why some methods perform better (L198-206), our work is primarily of exploratory and analytical nature (such as Recht et al. 2019, Koh et al. 2021). We bring to light that current DML benchmarks are insufficient by various metrics (fixed difficulty, semantically close to i.i.d, ... - see sections 1 \& 3 and L37-57), how performance differences between methods are increased when looking at more realistic OOD settings (Fig. 2), provide a new benchmark to evaluate DML methods across a spectrum of train-test shifts to simulate various realistic downstream scenarios (L90-103), highlight that metrics linked to generalization in DML do not hold when evaluated under such a comprehensive setting (Fig. 4) and that as a conceptual approach, few-shot DML, which has not been looked at before, can offer a disproportionate benefit when shifts are not known a priori and become increasingly more difficult.
> As such, our work both highlights the necessity for future, more principled research into the true (zero-shot) generalization capabilities of DML methods, as well as offering the test-bed on which this can be done.
>
> #### **Re. Introducing a simple method to mitigate the problem can be a plus to this paper:**
> As highlighted in our reply above, the primary target of our work was to pinpoint shortcomings of current evaluation protocols in DML, study current approaches under settings more reflective of realistic generalization tasks, and provide a new testbed and starting points for future, more principled research. However, we also introduce few-shot DML, which to the best of our knowledge has not been explored so far, as a general conceptual approach to *mitigate the problem* and improve OOD generalization performance, especially when encountered with more challenging shifts. See also our reply to reviewer VKkF.
>
> #### **Re. Importance of benchmark as machine learning does not aim to work under OOD shifts:**
> We believe that out-of-distribution generalization is crucial in a wide range of realistic application settings. This is further underlined by a large selection of publications (see e.g. Engstrom et al. 2017, Recht et al. 2019, Hendrycks \& Dietterich et al. 2019, Roth et al. 2020, Krueger et al. 2020, Koh et al. 2021) which investigate and showcase that for more realistic applications of machine learning methods, the assumption that test data will always be i.i.d. as the training data is too strong. And while the OOD shifts do not always have to be of semantic nature as is commonly the case in visual similarity applications found in DML (see Related Works, Section 2), such shifts are frequently encountered in practice (e.g. Krueger et al. 2020, Koh et al. 2021). Hence, conducting research on understanding how machine learning models can be improved under OOD settings is important as also pointed out by the other reviewers. To this end, we believe that our paper and novel benchmark ooDML will help and inspire future work.
>
> #### **Re. Relative ranking of methods:**
> The ranking in parts can change quite dramatically (see e.g. on SOP, L198-206), where previous SOTA methods actually perform much worse in more challenging settings.
> In addition, for benchmarks and methods where rankings are roughly retained, this also constitutes a novel, very valuable insight, highlighting that there are approaches to DML that retain benefits across OOD shifts.
> Finally, this benchmark is crucial in that it showcases that performance saturation between methods is more significant on the prevailing default, "close-to-iid" splits, but becomes less severe when looking at the spectrum of OOD shifts. While results on default split would suggest that the choice of method does not really matter much, our ooDML benchmark shows that conceptual differences indeed have an impact, especially if test shifts are not known *a priori*.
>
> #### **Re. Limitations \& Societal Impact:**
> We thank the reviewer for pointing this out, and have added a section highlighting both limitations and societal impact of our work.

---

### Decision · Program_Chairs · 2021-09-27

**Decision:**

Accept (Poster)

**Comment:**

This paper proposed a new benchmark for deep metric learning under varying degree of distribution shift. although the paper does not take a step in providing a solution, we are hopeful that this work inspires follow up work on OOD generalization. I request the authors to incorporate the reviews and discussions in the camera ready version.